# Usual prevention in unusual settings: A scoping review of place-based health interventions in public-facing businesses

Jack Tsai[1,2]*, Nicholas A. McCann[1]

**1** School of Public Health, University of Texas Health Science Center at Houston, Houston, Texas, United States of America, **2** Department of Psychiatry, Yale University School of Medicine, New Haven, Connecticut, United States of America

* Jack.Tsai@uth.tmc.edu

## Abstract

Place-based health interventions may help reach underserved populations. This scoping review summarizes the peer-reviewed literature on the type and effects of place-based health interventions in unconventional public-facing business settings (e.g., retail and services). A literature search was conducted in PubMed, Google Scholar, and APA PsycNet for studies from 1990–2023. Inclusion criteria for studies were: conducted in the United States, delivered a health intervention, based on an unconventional business setting, and targeted a specific health condition. An initial search yielded 2,727 unduplicated studies, which was filtered to 42 studies included in this review. These 42 studies were categorized based on health conditions of focus, including cardiovascular health (12 studies); HIV (6 studies); diabetes (5 studies); cancers (13 studies); and all other conditions (14 studies). The most common unconventional public-facing business settings for health interventions included barbershops or beauty salons; interventions included health education, preventative screenings, pharmacy services, and connections to local healthcare providers and resources. Notably, 34 (81%) of the studies targeted Black populations. Studies reported positive responses from participants for place-based interventions, increased awareness and screening of health conditions, more referrals to healthcare services, and improved health outcomes. While there have been 9 randomized trials conducted across various health conditions, these trials are limited to focus on a few select settings and lack of objective health outcome measures. These findings highlight the need for more rigorously designed studies in diverse settings that can effectively evaluate the impact of place-based health interventions. The existing literature suggests health interventions delivered in public-facing business settings may be a promising strategy to reach underserved populations.

## Introduction

An important part of public health is reaching all segments of the population where they live, work, and play [1]. However, there are various groups in the United States that can be

**Data Availability Statement:** All relevant data are within the manuscript. This is a review paper so there is no specific datasets. We do provide the

syntax for our literature search in the
Supplementary Information.

**Funding:** The author(s) received no specific
funding for this work.

**Competing interests:** The authors have declared
that no competing interests exist.

considered underserved, including some low-income, racial/ethnic minority, and immigrant populations. Thus, there have been increasing efforts to develop unique ways to outreach, engage, and serve these groups. Place-based interventions is one approach to reach people "where they are." Place-based interventions can range widely, from changing physical environments and areas where groups live (e.g., creating bike lanes to facilitate exercise) to increasing access to resources in neighborhoods (e.g., introducing new farmers markets or transport facilities).

A recent review examined 51 studies of locally-delivered place-based interventions across three elements of place and health: the physical, social, and economic environments [2]. However, these interventions were mostly focused on altering the "place" of people as the intervention, which can disconnect individuals from their familiar community contexts. In contrast, delivering traditional health interventions in unconventional settings—such as salons and barbershops—maintains the familiarity and trust that these spaces offer, potentially leading to better engagement and participation from target populations. For example, a number of novel studies have developed health promotion interventions in salons and barbershops to reach African American communities, which have been summarized in a recent systematic review [3]. However, the vast majority of health promotion activities in real-world settings and in the research literature continue to occur in traditional settings, such as primary care clinics, and clinics embedded in pharmacies, grocery stores, and community health centers.

The Coronavirus Disease-2019 (COVID-19) pandemic also helped popularize these types of place-based health interventions, which were often referred to as "pop-up clinics" by offering COVID-19 vaccines across many public settings, such as groceries, malls, schools, etc. [4]. But what types of other interventions and in which types of settings have these place-based interventions been used? Specifically, what unconventional public-facing business settings have been involved, and which communities have been served?

No prior review found has examined the full scope of these types of place-based health interventions delivered at less conventional public-facing business settings across all health conditions and racial/ethnic groups. Such a review would help public health practitioners, researchers, and business owners understand what is possible in terms of implementing interventions in public-facing business settings. In particular, a review that examines not only the types of settings, but the health conditions targeted and the outcomes can further knowledge about how to reach underserved and minority populations in these settings.

To address this knowledge gap, we conducted a scoping review of studies on delivering preventive health interventions in unconventional public-facing business settings, such as retail and service settings, to understand the type of interventions, the diversity of settings, the broad range of health conditions targeted, and how these interventions have been implemented across various racial and ethnic populations.

## Methods

### Protocol

A review protocol was developed based on the Preferred Reporting Items for Systematic Reviews and Meta-Analyses extension for Scoping Reviews (PRISMA-ScR) guidelines [5]. While this protocol was not registered, it followed a systematic approach aligned with PRISMA-ScR standards. The three major search databases used were PubMed, Google Scholar, and APA PsycNet. The search range in each database was set from 1960 to the present; however, the oldest study that met the inclusion criteria was published in 1995, and the most recent in 2023. Inclusion criteria were studies conducted in the United States, provided health interventions (e.g., education, disease screening, connection with healthcare providers, pharmacy

services) to clients in one or more business settings for one or more diseases or conditions, were written in English, and were peer-reviewed. Exclusion criteria were studies that were not peer-reviewed, not published in English-written journals, or did not report study of a health intervention in a business setting. We only included peer-reviewed studies to ensure the review focused on studies which had been previously assessed and deemed acceptable for publication as a quality control measure for this review.

## Search method

The two authors participated in all stages of the review, with one author leading study screening, selection, and data extraction and the other author assisting and double-checking this work for validity, and rerunning processes when there were discrepancies. Searches were initially conducted in PubMed using keywords and Boolean operators like "public health AND barbershop OR laundromat OR hair salon OR movie theater OR nail salon OR mechanic." The full syntax of keywords and strategies used in these initial searches are provided in S1 Table. The initial search returned a total of 2,727 unduplicated studies (PubMed 749, Google Scholar 1,930, APA PsycNet 48). Filtering and brief analysis of the titles and abstracts led to exclusion of 2,603 studies and a total of 124 studies entering the initial review process.

## Review process

Between January and April 2024, the two authors reviewed all 124 studies. To reduce potential bias, the two authors conducted a calibration exercise, independently reviewing a subset of studies to establish consistency before proceeding with selecting studies and extracting data from the selected studies. In total, 77 studies were removed for not meeting the inclusion criteria following an initial preliminary review of the studies by title and abstract (PubMed 76, Google Scholar 1). Twelve of the excluded studies were systematic reviews, and the other 65 studies were excluded due to conduction of the studies outside of the United States, providing health interventions outside of unusual business settings, or only surveying business clients without providing additional health intervention services. After a secondary review (NM), five more studies were removed after the full text articles were assessed for eligibility due to not meeting the inclusion criteria. This resulted in a final selection of 42 studies. These 42 studies were then sorted into three categories based on targeted health conditions: 1) cardiovascular health conditions, HIV, and diabetes (20 studies), 2) cancers (13 studies), and 3) all other conditions (14 studies). Note that certain included studies may fall into more than one category due to targeting multiple diseases or conditions through its intervention. Fig 1 includes a CONSORT diagram summarizing the study identification and review process.

## Data extraction and synthesis

Data extraction was conducted using an Excel-based form, which included fields for study design, business setting, health intervention, population of interest, sample size, target disease or condition, data findings, and conclusions. This Excel form served as a tracking and synthesis tool, while Zotero was used for citation management. To ensure consistency, data extraction was piloted by both reviewers on a subset of studies, following which any discrepancies were discussed and resolved. Synthesized data were categorized descriptively across intervention type, health condition, and setting, allowing for a narrative overview.

To assess the quality of each study included in this review, each study was evaluated using the NIH Study Quality Assessment Tools, matched to the appropriate study type [6]. Observational cohort and cross-sectional studies—including cross-sectional studies, longitudinal studies, descriptive studies, and cohort studies—were assessed using the *Quality Assessment Tool*

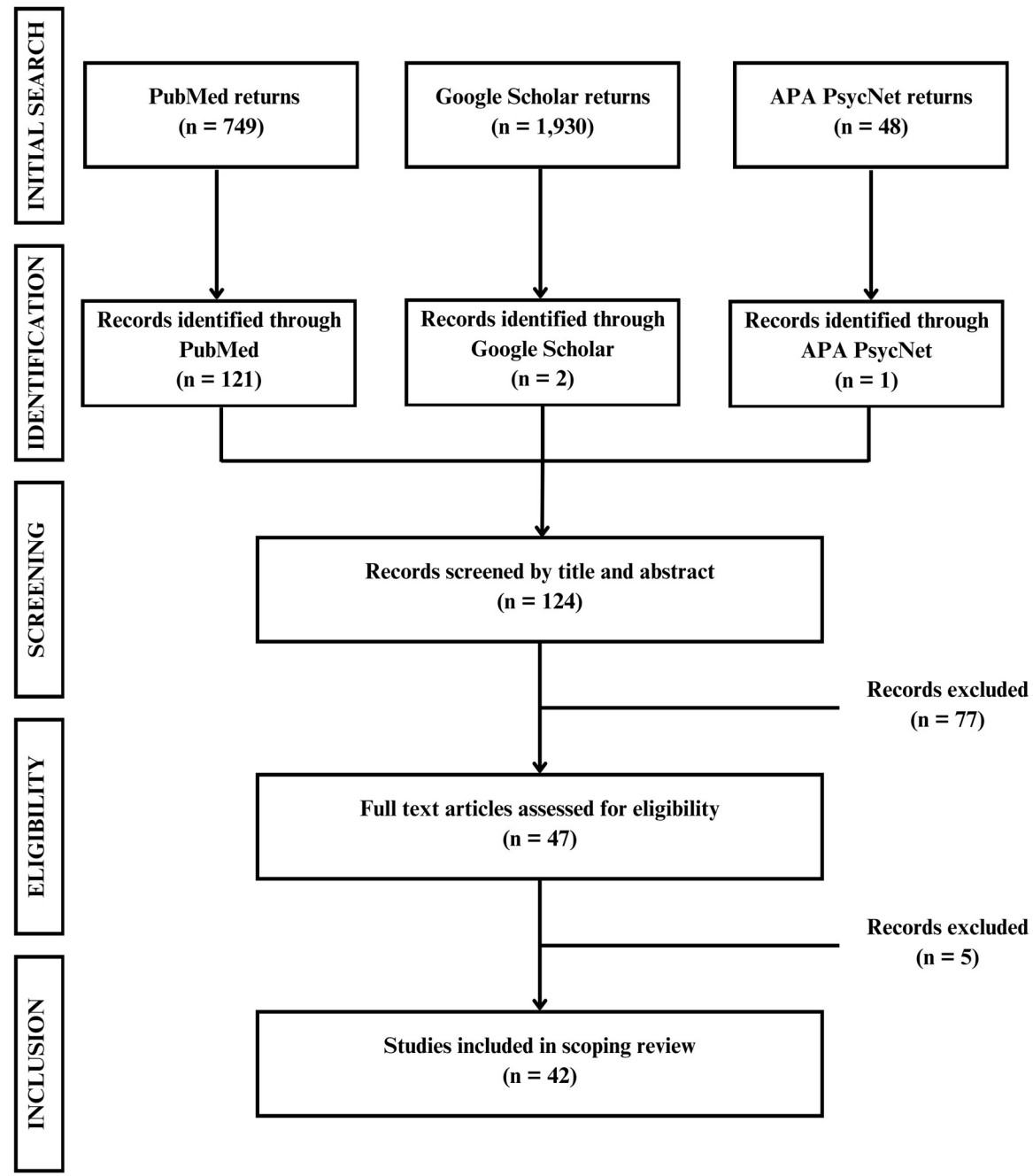

**Fig 1. Flow chart of study identification and review process.**

*for Observational Cohort and Cross-Sectional Studies.* Pre-post studies, such as feasibility studies, program evaluations, and pilot studies, were evaluated with the *Quality Assessment Tool for Before-After (Pre-Post) Studies With No Control Group.* Finally, controlled intervention studies, specifically randomized controlled trials (RCTs), were assessed using the *Quality Assessment of Controlled Intervention Studies.* By applying these tools, we ensured that each study design underwent a rigorous and relevant quality assessment. This assessment provided insight into the methodological rigor and potential risk of bias across the included studies.

## Results

The 42 studies identified were categorized by the health condition(s) they were focused on in Tables 1–3. Table 1 contained studies providing interventions for cardiovascular health conditions, HIV, and diabetes [7–26]. Table 2 contained studies providing interventions for different cancers [12, 27–38], and Table 3 contained studies providing interventions for all other conditions not falling into the previously listed categories [8, 12, 18, 39–48]. Each table lists the study author(s), study design, intervention setting (e.g., barbershop, nail salon, etc.), study population, sample size, health condition targeted by intervention, geographic area of study, major data findings, and overall findings. Note that some studies provided interventions for multiple conditions with certain studies falling into multiple tables [8, 12, 18]. Among the 42 studies, 36 studies involved health education as an intervention, 20 studies offered preventative screening as an intervention, 9 studies referred and facilitated connection to local healthcare providers and resources, and 3 studies offered pharmacy services as an intervention.

### Cardiovascular health conditions

A total of 12 studies were identified by their provided interventions for cardiovascular health conditions. Of these, 2 studies provided interventions for peripheral artery disease, 9 for hypertension or blood pressure monitoring, and 1 for heart disease.

Both studies investigating peripheral artery disease (PAD) at barbershops sampled Black men in a midwestern state, each with a sample size of 37 participants [7, 9]. One of these studies used a longitudinal study design [7], while the other was a qualitative analysis and was a sub-study of the longitudinal study [9]. In the longitudinal study [7], participants completed three visits to the barbershop: a first visit initially screened participants for PAD, a second visit rescreened participants for PAD and presented a PAD education video 4–6 weeks later, and a third visit in which exit interviews and assessments were conducted. The trial ultimately diagnosed PAD in 5/31 (16.1%) of participants and overall awareness of PAD began low at the beginning of the study but significantly increased between the initial and exit visit assessments.

The qualitative study [9] was conducted at the final exit assessment of the longitudinal study [7] and involved individual interviews to understand the perspectives of Black men in receiving barbershop-based screening and PAD education. Several common themes arose such as acknowledgement of barriers like fear, trust, and healthcare access, but participants indicated that the barbershop intervention enhanced knowledge of PAD and cardiovascular disease.

A total of 9 studies investigating hypertension or blood pressure management were identified, and all 9 studies targeted these conditions in either Black men or predominantly Black populations and offered interventions at barbershops or hair salons [8, 10, 11, 13–18]. These studies were conducted in all five regions of the United States (Northeast, Southwest, West, Southeast, and Midwest), and study designs included program evaluations, cluster-randomized trials, exploratory studies, longitudinal studies, pilot intervention studies, cohort studies, and descriptive studies. Additionally, the sample sizes of these hypertension intervention studies ranged widely from 10 to 14,000 participants.

Each of the 9 studies incorporated blood pressure screening into their interventions in different ways. For example, studies that investigated hypertension with larger sample sizes (i.e., 680–14,000) focused on taking blood pressure measurements, using self-report questions asking about recent blood pressure screenings, connecting participants to providers, or training barbers and hair stylists to serve as lay health educators [8, 16–18]. Overall, these larger studies found that barbershops and beauty salons served as suitable locations for blood pressure

**Table 1. Cardiovascular health, HIV, and diabetes.**

| Study | Study Design | Setting | Intervention | Study Population | Sample Size | Health Condition | Geographic Area | Data Findings | Main Conclusions Stated by Authors | Quality Rating[b] |
|---|---|---|---|---|---|---|---|---|---|---|
| White Solaru et al., 2022 | Longitudinal study | Barbershop | Peripheral artery disease screening and education | Black men | n = 37 participants | Peripheral artery disease (PAD) | Cleveland, OH | • PAD identified in 5/31 (16.1%) of participants • Low overall PAD awareness • Significant improvement in PAD awareness assessment scores obtained at the initial and exit visits (9.93±4.23 to 12.50±4.41, P = 0.004) | "Barbershop-based screening for PAD among Black men revealed a higher-than-expected PAD prevalence and low PAD awareness. An educational video was effective at increasing PAD awareness. Ankle-brachial index screening and educational outreach in the barbershop may be a feasible and effective tool to diagnose PAD and reduce PAD disparities among Black men at highest risk." | 9/14 |
| [a]Nadison et al., 2022 | Program Evaluation | Barbershops and beauty salons | Increase health awareness/ knowledge and reduce health disparities by increasing access to no-cost health care services | Predominantly Black populations | n = 1823 participants | Blood pressure, diabetes, tobacco, cholesterol, and social services (fitness, job search support, mental health, health insurance) | West Baltimore, Maryland | • 8000 clinical and social services provided between September 2016 and March 2020.—Blood pressure (n = 2317), diabetes (n = 469), tobacco (n = 448), and cholesterol (n = 443) were most accessed clinical screening services. • n = 2 median number of clinical services provided per client • Fitness (n = 1496), job search support (n = 1123), mental health (n = 603), and health insurance (n = 455) were most accessed social services | "The initiative delivered critical health and social support services through a partnership with an established integrated health care system, community barbershops and beauty salons, a mobile health team, and social supports. This novel program utilized a mobile health clinic to provide extensive clinical services complemented by on-site social services. Patterns of service utilization and lessons learned could inform the design of similar programs." | 6/12 |

*(Continued)*

**Table 1.** (Continued)

| Study | Study Design | Setting | Intervention | Study Population | Sample Size | Health Condition | Geographic Area | Data Findings | Main Conclusions Stated by Authors | Quality Rating[b] |
|---|---|---|---|---|---|---|---|---|---|---|
| Coy et al., 2023 | Qualitative analysis | Barbershop | Peripheral artery disease education | Black men | n = 37 participants | Peripheral artery disease (PAD) | Cleveland, OH | • 59.3 ± 11.2 years = mean age of participants<br>• 93% of participants resided in socioeconomically disadvantaged zip codes<br>• Increased awareness of PAD and acceptability of barbershop-based screenings for PAD, advocacy for systemic changes to improve the health of the community, and a desire among participants to increase knowledge about cardiovascular disease | "Participants were overwhelmingly accepting of PAD screenings and reported increased awareness of PAD and propensity to seek healthcare due to engagement in the study. Participants provided insight into barriers and facilitators of health and healthcare-seeking behavior, as well as into the community and the barbershop as an institution." | N/A |
| Victor, et al., 2019 | Cluster-randomized trial | Barbershop | Connection with pharmacists for BP medication prescriptions (intervention group) or BP education and promotion of follow-up with provider (control group) | Black men | n = 319 participants and n = 52 Los Angeles County barbershops | Hypertension | Los Angeles County | • Baseline, mean systolic BP was 152.4 mm Hg in the intervention group and 154.6 mm Hg in the control group<br>• Mean systolic BP fell by 28.6 mm Hg (to 123.8 mm Hg) in the intervention group and by 7.2 mm Hg (to 147.4 mm Hg) in the control group after 12 months<br>• Mean reduction was 20.8 mm Hg greater in the intervention (95% CI, 13.9–27.7; P<0.0001)<br>• BP <130/80 mm Hg was achieved by 68.0% of the intervention group versus 11.0% of the control group (P<0.02)—Cohort retention at 12 months was 90% in both groups. | "Among Black male barbershop patrons with uncontrolled hypertension, health promotion by barbers resulted in large and sustained BP reduction over 12 months when coupled with medication management by American Society of Hypertension-certified pharmacists." | 11/14 |

(Continued)

**Table 1.** (Continued)

| Study | Study Design | Setting | Intervention | Study Population | Sample Size | Health Condition | Geographic Area | Data Findings | Main Conclusions Stated by Authors | Quality Rating[b] |
|---|---|---|---|---|---|---|---|---|---|---|
| Blyler et al., 2021 | Pilot Study | Barbershops & Telehealth | Investigated effectiveness of telehealth appointments following improved BP control after hypertension diagnosis and prescription from pharmacist at barbershop | Black men | n = 10 participants | Hypertension | Los Angeles, CA | • Baseline BP of 155±14/ 83.9±11 mmHg decreased by -28.7±13/-8.9±15 mm Hg (P<0.0001) • Hypertension control (≤130/80 mm Hg) was 67% (6 of 9), numerically greater than the 63% observed in LABBPS • Mean number of in-person visits decreased from 11 in LABBPS to 6.6 visits over 12 months | "Virtual visits represent a viable substitute for in-person visits, both improving pharmacist efficiency and reducing cost while preserving intervention potency." | 8/12 |
| [a]Waters et al., 2023 | Longitudinal study | Barbershop | Diabetes, heart disease, stroke, colon cancer, and breast cancer screening and education | At risk participants from community locations | n = 356 participants | Diabetes, heart disease, stroke, colon cancer, and breast cancer | Unknown | • 18% of participants believed their diabetes risk was lower than the information provided • 40% believed their risk was higher • 42% accepted the information. | "There are likely multiple cognitive, affective, and motivational explanations for risk skepticism. Understanding these explanations and developing interventions that address them will increase the effectiveness of precision medicine and facilitate its widespread implementation." | 11/14 |
| Hess et al., 2007 | Longitudinal study | Barbershop | Monitored hypertension treatment and blood pressures of participants by researchers (study 1) and barbers (study 2) | Black men | n = 36 for enhanced intervention group, n = 27 for contemporaneous comparison group (Study 1), and n = 321 for Study 2 | Hypertension | Dallas County, TX | • Study 1: BP fell by 16 +/-3/9+/-2 mm Hg in the enhanced intervention group but was unchanged in the comparison group (P<0.0001, adjusted for age and body mass index). HTN treatment and control increased from 47% to 92% (P<0.001) and 19% to 58% (P<0.001), respectively, in the enhanced intervention group, whereas both remained unchanged in the comparison group. • Study 2: Six barbers recorded 8953 BP checks during 11,066 haircuts. Treatment and control increased progressively with increasing intervention exposure (P<0.01) among HTN participants. | "Taken together, these data suggest that Black-owned barbershops can be transformed into effective HTN detection, referral, and follow-up centers." | 12/14 |

(Continued)

**Table 1.** (Continued)

| Study | Study Design | Setting | Intervention | Study Population | Sample Size | Health Condition | Geographic Area | Data Findings | Main Conclusions Stated by Authors | Quality Rating[b] |
|---|---|---|---|---|---|---|---|---|---|---|
| Victor et al., 2011 | Cluster-randomized trial | Barbershop | Blood pressure screening, hypertension education materials, lay health education by barbers, and connection of participants to physicians | Black men | n = 8 shops with 77 patrons per shop for comparison group, and n = 9 shops with 75 patrons per shop for intervention group | Hypertension | Dallas County, TX | • HTN control rate increased more in intervention barbershops than in comparison barbershops (absolute group difference, 8.8% [95% confidence interval (CI), 0.8%-16.9%]) (P = .04)<br>• Marginal intervention effect was found for systolic BP change (absolute group difference, -2.5 mm Hg [95% CI, -5.3 to 0.3 mm Hg]) (P = .08) | "The effect of BP screening on HTN control among Black male barbershop patrons was improved when barbers were enabled to become health educators, monitor BP, and promote physician follow-up." | 10/14 |
| Rader et al., 2013 | Cluster-randomized trial | Barbershop | Investigated differential treatment of hypertension by primary care providers and hypertension specialists with barbers as education mediators | Black men | Comparison-arm patrons (n = 68) treated by PCPs with (1) intervention-arm patrons (n = 37) treated by PCPs or (2) intervention-arm patrons (n = 33) who lacked access to PCPs and were treated by HTN specialist physicians serving as safety net providers | Hypertension | Dallas County, TX | • HTN specialist group had higher baseline systolic BP than the others (162 ± 3 vs 155 ± 2 and 154 ± 2 mmHg, respectively; p <0.01)<br>• Systolic BP reduction was 21 ± 4 mmHg greater than in the comparison group (p <0.0001), when barbers referred patrons to hypertension specialists but was no different when they referred to PCPs (4 ± 4 mm Hg, p = 0.31)—Specialist-treated patrons received more BP medication and different classes of medication than PCP-treated patrons | "In conclusion, the barber-based intervention-if connected directly to specialty-level medical care-could have a large public health impact on hypertensive disease in Black men." | 10/14 |
| Ferdinand, 1995 | Pilot Study | Barbershops and Beauty Salons | Blood pressure screening and education | Predominantly Black populations | n = 1350 clients | Hypertension | New Orleans, LA | • n = 877 blood pressure checks<br>• Approximately 32.4% of African American participants have hypertension with systolic or diastolic pressure of 140/90<br>• n = 21 participants were referred to physicians for immediate care within 1 week due to dangerously elevated blood pressures. | Increased blood pressure screenings in barbershops and beauty salons. | 6/12 |

(*Continued*)

**Table 1.** (Continued)

| Study | Study Design | Setting | Intervention | Study Population | Sample Size | Health Condition | Geographic Area | Data Findings | Main Conclusions Stated by Authors | Quality Rating[b] |
|---|---|---|---|---|---|---|---|---|---|---|
| Mendy et al., 2014 | Descriptive study | Barbershop | Blood pressure screening and education | Black men | n = 14 barbershops and n = 686 participants | Hypertension | Mississippi Delta region | • 14.7% had normal blood pressure • 48.5% had prehypertension • 36.4% had high blood pressure—35% reported having a personal doctor • 43% reported having health insurance • Of the men screened, 34.3% were referred to a health care provider for follow up | "MDHC aims to increase the number of barbers engaged in blood pressure screenings and the number of participants screened at each barbershop. In addition, MDHC plans to establish a provider network to better link participants with high blood pressure to local health care providers." | 7/14 |
| [a]Madigan et al., 2007 | Cohort study | Barbershops and beauty salons | Chronic kidney disease, hypertension, and diabetes education and connection to resources | Predominantly Black populations | n = 700 trained stylists and n = 14,000 clients | Chronic kidney disease, hypertension, and diabetes | Michigan | • 60% of clients indicating that they have taken steps to prevent diabetes, hypertension, and chronic kidney disease or to seek a physician's advice | "With nearly 60% of clients indicating that they have taken steps to prevent diabetes, hypertension, and chronic kidney disease or to seek a physician's advice, the Healthy Hair program appears to be effective in the short term in prompting attention to healthy behaviors and increasing risk awareness." | 10/14 |

(*Continued*)

**Table 1.** (Continued)

| Study | Study Design | Setting | Intervention | Study Population | Sample Size | Health Condition | Geographic Area | Data Findings | Main Conclusions Stated by Authors | Quality Rating[b] |
|---|---|---|---|---|---|---|---|---|---|---|
| Wilson et al., 2014 | Longitudinal study | Barbershop | HIV education | Black men | n = 80 men recruited, n = 78 men completed program, and n = 71 completed 3-month assessment | HIV | New York City, NY | • Proportion of men who reported not having engaged in any unprotected sex in the past 3 months increased from the baseline to the follow-up administration (25% to 41%, p = 0.007). • Proportion of men who reported having unprotected sex with two or more women in the past 3 months declined from baseline to follow-up (46% to 17%, p = 0.0001). • Proportion of men reporting favorable attitudes toward condoms and confidence in their self-efficacy to use condoms consistently increased from baseline to 3-month follow-up (p < 0.05) • BTWB participants reported increased perceptions of community empowerment (p = 0.06) from baseline to follow-up, but no differences were detected in HIV stigma (p = 0.11) | "Specifically, attitudes and self-efficacy toward consistent condom use improved, and respondents reported lower levels of sexual risk behavior from baseline to follow-up (all p < 0.05). Perceptions of community empowerment also increased (p = 0.06)." | 10/14 |

(*Continued*)

**Table 1.** (Continued)

| Study | Study Design | Setting | Intervention | Study Population | Sample Size | Health Condition | Geographic Area | Data Findings | Main Conclusions Stated by Authors | Quality Rating[b] |
|-------|-------------|---------|--------------|------------------|-------------|------------------|-----------------|---------------|-----------------------------------|------------------|
| Dale et al., 2023 | Descriptive study | Corner stores, beauty supply stores, laundromats, mechanics, barbershops | HIV prevention and education | Predominantly Black populations | n = 13 community businesses, n = 5 health organizations, n = 677 community residents | HIV | Miami, FL | • n = 12,434 condoms distributed<br>• n = 131 voluntary HIV tests completed<br>• 68.8% never heard of PrEP, 8% no HIV testing ever, 65.9% no primary care provider<br>• Positive feedback from residents (e.g., 70% very satisfied, 21% satisfied; 62% strongly agree and 25% agree they would participate again)<br>• Corner/food store events had the highest average for residents who completed surveys and testing per event (averaged 93 surveys and 18 tests across 4 events), followed by the laundromat (averaged 49 surveys and 19 tests for 1 event), barber/salon/beauty (averaged 42 surveys and 9 tests across 3 events), and the other category (averaged 22 surveys and 2 tests) | "The FPI bundled implementation strategy shows promise to deliver health prevention/intervention for HIV and other health conditions to communities facing health inequities and for whom the current system for delivering care is insufficient." | 8/14 |
| Wilson et al., 2019 | Cluster-randomized trial | Barbershop | HIV prevention and education | Black men | n = 53 barbershops (24 intervention, 29 control) and n = 860 men (436 intervention, 424 control) | HIV | Brooklyn, NY | "Intervention exposure was associated with a greater likelihood of no condomless sex (64.4%) than control group participation (54.1%; adjusted odds ratio = 1.61; 95% confidence interval = 1.05, 2.47)." | "Program exposure resulted in reduced sexual risk behaviors, and the program was acceptable for administration in partnership with barbershops." | 12/14 |

*(Continued)*

**Table 1.** (Continued)

| Study | Study Design | Setting | Intervention | Study Population | Sample Size | Health Condition | Geographic Area | Data Findings | Main Conclusions Stated by Authors | Quality Rating[b] |
|---|---|---|---|---|---|---|---|---|---|---|
| Jemmott et al., 2017 | Qualitative analysis | Barbershop | Knowledge of HIV and safe sex practices were screened in barbershops. Then, HIV and safe sex education videos were played for participants on iPads during haircuts. | Black men | n = 14 barbers, n = 5 barbershop owners, n = 48 cross-sectional survey participants, n = 4 participants in iPad testing | HIV | Philadelphia, PA | "Mean ratings indicated that participants liked and learned a lot from the video, activities, and homework assignments. Mean scores also indicated that participants felt fairly comfortable talking and sharing their thoughts and performing the exercises and role-play. Participants' ratings of the facilitators were also very high. Participants liked the facilitators and felt that facilitators were knowledgeable about the intervention content and were good role models. Mean ratings also indicated that participants would recommend the program to other African American men." | "The results of this study suggest that the development of a gender-specific culturally tailored HIV risk reduction intervention may a promising strategy to reducing the transmission of HIV and STIs among young heterosexual African American men." | N/A |
| Hawkins et al., 2022 | Program Evaluation | Barbershops and Beauty Salons | HIV screening and education in barbershops | Racial and ethnic minorities | n = 32 salons, n = 1,124 participants | HIV | Delaware | • n = 6,000 people reached<br>• n = 1,124 HIV tests administered<br>• n = 0 positive test results thus far | "As a result of the messages given by POLs, AIDS Delaware has seen a dramatic increase in HIV testing numbers." | 6/12 |
| Jemmott et al., 2023 | Cluster randomized controlled trial | Barbershop | HIV/STI risk-reduction intervention based on the theory of planned behavior and formative research or an attention-matched violence-prevention control intervention | Black men | n = 24 matched pairs of barbershops, n = 618 men (319 in the HIV/STI intervention and 299 in the control intervention) | HIV | Philadelphia, PA | • "Generalized estimating equation analysis indicated that the direct effect of the HIV/STI intervention in increasing consistent condom use postintervention was nonsignificant (odds ratio = 1.13, 95% confidence interval: 0.73–1.75), adjusting for clustering among participants in barbershops and baseline condom use."<br>• "Mediation analysis using the product-of-coefficients approach revealed indirect effects of the intervention." | "Sexual risks among young African-American men can be reduced by barber-led theory-based, culturally appropriate HIV/STI risk-reduction interventions in barbershops in high HIV prevalence neighborhoods that increase behavioral beliefs and self-efficacy." | 14/14 |

(*Continued*)

**Table 1.** (Continued)

| Study | Study Design | Setting | Intervention | Study Population | Sample Size | Health Condition | Geographic Area | Data Findings | Main Conclusions Stated by Authors | Quality Rating[b] |
|---|---|---|---|---|---|---|---|---|---|---|
| Balls-Berry et al., 2015 | Focus Group Study/ Qualitative research | Barbershop | Diabetes education and prevention | Black men | n = 13 participants (Group 1, n = 6; Group 2, n = 7) | Diabetes | Rochester, Minnesota | "Participants identified diet and exercise as essential components of diabetes prevention. Additionally, participants mentioned that family history contributes to diabetes. Participants agreed that barbershops are an appropriate setting for data collection and health education on diabetes for Black men." | "Findings indicate that Black men are generally aware of diabetes. The community-engaged research process allowed for development of a culturally appropriate research study on diabetes. This study is the foundation for developing a culturally appropriate health education program on diabetes for Black men." | N/A |
| Osorio et al., 2020 | Cross-sectional study | Barbershop | Hemoglobin A1C testing in barbershops and corresponding diabetes education for positive screens | Black men | n = 290 participants | Diabetes | Brooklyn, NY | • Of 290 participants, 26 (9.0%) had an HbA1c level of 6.5% or higher and 3 (1.0%) had an HbA1c level of 7.5% or higher.<br>• The highest HbA1c level was 7.8%.<br>• 82 participants (28.3%) had an HbA1c level between 5.7% and 6.4% (prediabetes).<br>• Of the 26 participants with undiagnosed diabetes, 16 (61.5%) were obese.<br>• Median age of these men diagnosed with diabetes was 41 (range, 22–65) years and 11 (42.3%) had an education of high school or less. | "Our findings suggest that community-based diabetes screening in barbershops owned by Black individuals may play a role in the timely diagnosis of diabetes and may help to identify Black men who need appropriate care for their newly diagnosed diabetes." | 7/14 |

[a]indicates study is present across multiple tables (Tables 1–3)

[b]Quality assessment of each study design was conducted using the NIH Study Quality Assessment Tools. Full titles of the tools used include: Quality Assessment of Controlled Intervention Studies, Quality Assessment Tool for Observational Cohort and Cross-Sectional Studies, and Quality Assessment Tool for Before-After (Pre-Post) Studies With No Control Group [6].

**Table 2. Cancers.**

| Study | Study Design | Setting | Intervention | Study Population | Sample Size | Health Condition | Geographic Area | Data Findings | Main Conclusions Stated by Authors | Quality Rating [b] |
|---|---|---|---|---|---|---|---|---|---|---|
| Luque et al., 2011 | Pilot study | Barbershop | Distribution of customized prostate cancer education materials by barbers | Black men | n = 8 barbers, n = 40 clients | Prostate cancer | Tampa, FL | • Significant increase in barbershop clients' self-reported knowledge of prostate cancer and in the likelihood of discussing PCS with a health care provider (p < .001). • All clients surveyed reported positively on the contents of the educational materials, and more than half (53%) had discussed prostate cancer at least twice with their barber in the last month. | "Based on the pilot results, the barber-administered intervention was an appropriate and viable communication channel for promoting CaP knowledge and awareness in a priority population, African American men." | 7/12 |
| [a]Waters et al., 2023 | Longitudinal study | Barbershop | Diabetes, heart disease, stroke, colon cancer, and breast cancer screening and education | At risk participants from community locations | n = 356 participants | Diabetes, heart disease, stroke, colon cancer, and breast cancer | Unknown | • 18% of participants believed their diabetes risk was lower than the information provided • 40% believed their risk was higher • 42% accepted the information. | "There are likely multiple cognitive, affective, and motivational explanations for risk skepticism. Understanding these explanations and developing interventions that address them will increase the effectiveness of precision medicine and facilitate its widespread implementation." | 11/14 |
| Sizer et al., 2022 | Qualitative analysis | Barbershop | Colorectal cancer education by barbers trained by nurse practitioners and colonoscopy referrals | Black men | n = 13 participants | Colorectal cancer | Rochester, New York | • n = 13 participants agreed to participate in the project • n = 9 participants scheduled colorectal cancer screening • n = 7 completed colonoscopy evaluation | "Barbers and nurse practitioners are an ideal partnership when seeking to disrupt the CRC health care disparity. Members of the Black community who may not routinely participate in preventive care can be innovatively educated to improve their health status." | N/A |

*(Continued)*

**Table 2.** (Continued)

| Study | Study Design | Setting | Intervention | Study Population | Sample Size | Health Condition | Geographic Area | Data Findings | Main Conclusions Stated by Authors | Quality Rating [b] |
|---|---|---|---|---|---|---|---|---|---|---|
| Luque et al., 2010 | Pilot study | Barbershop | Prostate cancer education by barbers and distribution of prostate cancer education materials | Black men | n = 8 barbers, n = 115 clients | Prostate cancer | Tampa, FL | • Training workshops led to a significant increase in mean prostate cancer knowledge scores among the barbers (60% before vs. 79% after; P < 0.05). • Barbers completed 115 encounter notecards and distributed approximately 500 brochures over two months. • Barbers were most likely to use the brochures (74%) in their educational encounters, and the topics most frequently discussed were prostate cancer (84%) and the PSA test (65%) • n = 9 referrals were documented on the notecards to community health agencies for under- and uninsured individuals for health screenings (prostate cancer checks). | "Training barbers to deliver a prostate cancer educational intervention is a feasible strategy for raising prostate cancer awareness of the disease among a priority population." | 8/12 |
| Cole et al., 2017 | Randomized controlled trial | Barbershop & Telehealth | Recruitment from barbershops and connected to resources for colorectal cancer screenings | Black men | n = 731 participants NOTE: "We randomized participants to 1 of 3 groups: (1) patient navigation by a community health worker for CRC screening (PN), (2) motivational interviewing for blood pressure control by a trained counselor (MINT), or (3) both interventions (PLUS)." | Colorectal cancer | New York City, NY | • Participants in the navigation interventions were significantly more likely than those in the MINT-only group to be screened for CRC during the 6-month study period (17.5% of participants in PN, 17.8% in PLUS, 8.4% in MINT; P < .01). | "Telephone-based preclinical patient navigation has the potential to be effective for older Black men. Our results indicate the importance of community-based health interventions for improving health among minority men." | 8/14 |
| Tingen et al., 1998 | Descriptive study | Barber shops, churches, industries, meal sites, car dealerships, civic organizations, and housing projects | Prostate cancer screening and education | Men | n = 1,552 participants | Prostate cancer | Southeastern state | • Predictors of participation in free prostate cancer screening were these: perceived benefits, being white, having at least a high school education, being married, and receiving the client navigator or combination educational intervention. • The Benefits Scale was significant (p = 0.013, odds ratio (OR) = 1.059) as a predictor for participation in screening when all demographic variables and educational interventions were controlled. | "The current research revealed that married men who perceive the benefits of participating in free prostate cancer screening were more likely to participate in screening." | 9/14 |

*(Continued)*

**Table 2.** (Continued)

| Study | Study Design | Setting | Intervention | Study Population | Sample Size | Health Condition | Geographic Area | Data Findings | Main Conclusions Stated by Authors | Quality Rating [b] |
|---|---|---|---|---|---|---|---|---|---|---|
| Kreuter et al., 2008 | Descriptive study | Beauty salons, churches, neighborhood health centers, laundromats, social service agencies, health fairs, and public libraries | Use of kiosks for breast cancer education and connection to local resources | Black women | n = 10,306 kiosk uses, n = 92 beauty salons, churches, neighborhood health centers, laundromats, social service agencies, health fairs, and public libraries | Breast cancer | Unknown | "Of the seven settings, only laundromats were found to provide both high reach (ie, frequent kiosk use) and high specificity (ie, a large proportion of users with no health insurance, unaware of where to get a mammogram, reporting no recent mammogram and barriers to getting one, and having little knowledge about breast cancer and mammography)." | "Systematic, data-based evaluations of potential dissemination channels can help identify optimal settings for cancer control interventions." | 9/14 |
| Kreuter et al., 2006 | Descriptive study | Beauty salons, churches, health fairs, neighborhood health centers, laundromats, public libraries and social service agencies. | Individually tailored education via magazines on breast cancer and mammography | Black women | n = 4,527 kiosk uses, n = 40 different host sites | Breast cancer | St. Louis, MO | "Highly significant differences among community settings were found in rates and patterns of kiosk use as well as user characteristics, breast cancer knowledge, and use of mammography." | "Findings inform strategic decision making about technology dissemination and community outreach to women needing information about breast cancer and mammography." | 10/14 |
| Wilson et al., 2008 | Randomized-controlled trial | Beauty salons | To assess the effectiveness of breast health promoting messages administered by salon stylists to clients in the salon settings | Black women | n = 40 salons, n = 1,185 pre-intervention surveys by salon clients, n = 1,210 assessments of clients were conducted | Breast cancer | Unknown | • 10% of women at control salons reported exposure to breast health messages, as opposed to 37% at experimental salons (OR 5.4, 95% CI 3.7–7.9)—Self-reported exposure to stylist-delivered messages was associated with improved breast self-examination rates (OR 1.6, 95% CI 1.2–2.1) and with greater intentions to have a clinical breast examination (OR 1.9, 95% CI 1.1–3.3) | "Hair salons are a potentially important venue for promotion of health behaviors related to breast cancer detection." | 6/14 |

*(Continued)*

**Table 2.** (Continued)

| Study | Study Design | Setting | Intervention | Study Population | Sample Size | Health Condition | Geographic Area | Data Findings | Main Conclusions Stated by Authors | Quality Rating [b] |
|---|---|---|---|---|---|---|---|---|---|---|
| Frencher et al., 2016 | Program Evaluation | Barbershop | Prostate cancer education and resources involving use of decision support instruments | Black men | n = 120 participants | Prostate cancer | Los Angeles, CA | • Culturally tailored decision support instruments (DSI) demonstrated a statistically significant increase in intention to screen. • Participants' degree of certainty in their decision-making process with regard to CaP screening increased following the culturally tailored DSI (p < .001). • Majority of participants planned on discussing CaP screening with a healthcare provider upon completion of the study. • At 3 months follow-up, half (n = 58) of the participants underwent PSA testing, which led to the diagnosis of CaP in one participant. | "Community-led interventions for CaP, such as cluster-randomized designs in barbershops, are needed to better assess the efficacy of DSI in community settings." | 6/12 |
| Sadler et al., 2011 | Cluster randomized trial | Beauty salons | Breast cancer education by cosmetologists and distribution of breast cancer education materials | Black women | n = 984 participants | Breast cancer | San Diego, CA | • 57% of women reported that health education materials were displayed in their salon • 80% of participants thought cosmetologists could effectively pass along health information to their client | "This intervention was well received by the participants and their cosmetologists and did not interfere with or prolong the client's salon visit. Women in the intervention group reported significantly higher rates of mammography compared to women in the control group. Training a single educator proved sufficient to permeate the entire salon with the health message, and salon clients agreed that cosmetologists could become effective health educators." | 7/14 |
| Sadler et al., 2000 | Pilot study | Beauty salons | Breast cancer education | Black women | n = 8 cosmetologists (participant sample size not specified) | Breast cancer | Unknown | "Both cosmetologists and clients found this an acceptable intervention. Nearly all women in the study demonstrated that they had heard the mainstream messages about the value of breast cancer early detection, but a considerable proportion appeared not to realize breast cancer's high level of morbidity and mortality within their own community." | "The results suggest this approach is worthy of further evaluation. | 6/12 |

(Continued)

**Table 2.** (Continued)

| Study | Study Design | Setting | Intervention | Study Population | Sample Size | Health Condition | Geographic Area | Data Findings | Main Conclusions Stated by Authors | Quality Rating[b] |
|-------|--------------|---------|--------------|------------------|-------------|------------------|-----------------|---------------|-----------------------------------|-------------------|
| Liman et al., 2005 | Pilot study | Beauty salons | Breast cancer education and prevention | Women | n = 162 participants | Breast cancer | North Carolina | • Most customers who responded in both salons (African American 83.3%, White 89.3%) reported talking with their cosmetologist about the BEAUTY Project during their salon visit. • More African American (71.2%) than White customers (53.8%) discussed the cancer prevention health brochures with their cosmetologist. • Customer responses to an open-ended question on what they liked "most" about the intervention listed: information about weight control and exercise to prevent cancer and the educational health display. • Approximately 70% of customers encouraged the continuation of the BEAUTY Project in other salons. | "Trained stylists reported they would continue delivering health messages after the 7-week pilot was completed; 81% of customers read the educational displays, and 86% of customers talked with their cosmetologist about the Bringing Education and Understanding to You Project. At 12 months, 55% of customers reported making changes in their health because of the conversations they had with their cosmetologist." | 7/12 |

[a]indicates study is present across multiple tables (Tables 1–3)

[b]Quality assessment of each study design was conducted using the NIH Study Quality Assessment Tools. Full titles of the tools used include: Quality Assessment of Controlled Intervention Studies, Quality Assessment Tool for Observational Cohort and Cross-Sectional Studies, and Quality Assessment Tool for Before-After (Pre-Post) Studies With No Control Group [6]

**Table 3. Other health conditions.**

| Study | Study Design | Setting | Intervention | Study Population | Sample Size | Health Condition | Geographic Area | Data Findings | Main Conclusions Stated by Authors | Quality Rating [b] |
|---|---|---|---|---|---|---|---|---|---|---|
| [a]Nadison et al., 2022 | Program Evaluation | Barbershops and beauty salons | Increase health awareness/knowledge and reduce health disparities by increasing access to no-cost health care services | Predominantly Black populations | n = 1823 participants | Blood pressure, diabetes, tobacco, cholesterol, and social services (fitness, job search support, mental health, health insurance) | West Baltimore, Maryland | • 8000 clinical and social services provided between September 2016 and March 2020.—Blood pressure (n = 2317), diabetes (n = 469), tobacco (n = 448), and cholesterol (n = 443) were most accessed clinical screening services. • n = 2 median number of clinical services provided per client • Fitness (n = 1496), job search support (n = 1123), mental health (n = 603), and health insurance (n = 455) were most accessed social services | "The initiative delivered critical health and social support services through a partnership with an established integrated health care system, community barbershops and beauty salons, a mobile health team, and social supports. This novel program utilized a mobile health clinic to provide extensive clinical services complemented by on-site social services. Patterns of service utilization and lessons learned could inform the design of similar programs." | 6/12 |
| Williams et al., 2020 | Cross-sectional study | Barbershops | Nutrition education and promotion (fruit and vegetable consumption) | Black men | n = 134 participants | Nutritional deficiencies | Jackson, Mississippi | • Mean total number of fruits and vegetables consumed by participants within 24 hours of the taking the survey was 1.63 (SD = 1.47). • Mean intention to initiate consuming 5 or more cups of fruits and vegetables per day score was 2.13 (SD = 1.17) • Practice for change (β = 0.462, P<0.001) and emotional transformation (β = 0.215, P<0.0001) accounted for 37.5% of the variance in the intention to sustain fruits and vegetables consumption behavior. | "Based on data found in the study, MTM appears to predict the intention to initiate and sustain fruit and vegetable intake of African American men." | 10/14 |
| [a]Waters et al., 2023 | Longitudinal study | Barbershop | Diabetes, heart disease, stroke, colon cancer, and breast cancer screening and education | At risk participants from community locations | n = 356 participants | Diabetes, heart disease, stroke, colon cancer, and breast cancer | Unknown | • 18% of participants believed their diabetes risk was lower than the information provided • 40% believed their risk was higher • 42% accepted the information. | "There are likely multiple cognitive, affective, and motivational explanations for risk skepticism. Understanding these explanations and developing interventions that address them will increase the effectiveness of precision medicine and facilitate its widespread implementation." | 11/14 |

*(Continued)*

**Table 3.** (Continued)

| Study | Study Design | Setting | Intervention | Study Population | Sample Size | Health Condition | Geographic Area | Data Findings | Main Conclusions Stated by Authors | Quality Rating[b] |
|---|---|---|---|---|---|---|---|---|---|---|
| Roberts-Dobie et al., 2018 | Descriptive study | Beauty salons | Trained hair stylists as lay health educators to increase knowledge about contraceptives and inform clients about financial support for long-acting reversible contraceptives at local family planning clinics | Women | n = 126 hairstylists, n = 177 participants | Unintended pregnancy | Midwestern states | • 60.4% (n = 107) indicated that they had discussed either contraceptives or preventing unintended pregnancies with their stylist. • n = 52 participants (29.4%) reported that they had talked to their health care provider about the contraceptive information they discussed with their stylist or received at the salon. • Four out of seven involved counties saw declines in unintended pregnancies between 3.6% and 10.7% and three saw increases between 1.3% and 7.8%, all with fluctuations over the 4-year period. | "Results from a subsample of participants who completed an online questionnaire (n = 177) indicate that hair stylists are a feasible method to link target populations to health information and to the health care system." | 7/14 |
| DiVietro et al., 2016 | Descriptive study | Hair salons | Intimate partner violence (IPV) screening | Women | Unknown | Intimate partner violence (IPV) | Connecticut | "Overall, reported past-year prevalence of physical abuse was 3.6%, past-year prevalence of sexual abuse was 2.7%, lifetime prevalence of emotional or physical abuse was 34.2%, and 5.3% of the sample reported that they had been hurt that day by their current or former partner. Past-year physical abuse was more common among women 30 years to 39 years old (9.1%), Black (9%), and single women (7.5%). Past-year sexual abuse was more common among women 20 years to 29 years old (13.8%), other races (6.7%), and single women (5.4%). Lifetime abuse was more common among women 50 years to 59 years old (13.8%), Black (36.1%), and divorced women (69.7%). Hurt-today abuse was more common among women younger than 20 years (12.5%), other races (13.3%), and women in common law relationships (25%)." | "Women in our study reported IPV prevalence rates consistent with national data. Documentation of IPV prevalence in hair salons will provide much-needed support for novel interventions such as CUT IT OUT, a national program designed to train hair stylists on how to recognize and refer IPV victims." | 6/14 |

(*Continued*)

**Table 3.** (Continued)

| Study | Study Design | Setting | Intervention | Study Population | Sample Size | Health Condition | Geographic Area | Data Findings | Main Conclusions Stated by Authors | Quality Rating[b] |
|-------|-------------|---------|-------------|-----------------|-------------|------------------|-----------------|---------------|-----------------------------------|-------------------|
| Beebe et al., 2018 | Feasibility study | Hair salons | Intimate partner violence (IPV) screening | Women | n = 203 participants | Intimate partner violence (IPV) | Connecticut | • n = 40/203 participants (20%) had experienced IPV in her lifetime. | "In identifying the prevalence of IPV within the salon setting, this study provides support for community-based programs and supports their legitimacy as an important locus for identifying women experiencing IPV and connecting them to resources." | 7/12 |
| Peddecord et al., 2008 | Descriptive study | Movie theaters | Influenza vaccination education and promotion before pre-film trailers | Adults with children 6 months to 2 years of age and adults over 50 years of age | n = 530 participants | Influenza | San Diego, CA | "Overall, 88% of exposed patrons reported seeing some type of movie ad. Among those who recalled any ad, 24% recalled the flu advertisement. In contrast, recall of flu-related news coverage was high, with over 95% of exposed and comparison interviewees recalling news stories during the campaign period. While 56% of those interviewed remembered one or more specific flu-related news items, individuals within this group who also had also been exposed to the movie ads were not more likely to recall flu campaign advertisements." | "Further research that compares movie ads with public service announcements (PSAs) in other venues is necessary to solidify our conclusions that movie advertising is a highly cost-effective medium for health communication." | 8/14 |

*(Continued)*

**Table 3.** (Continued)

| Study | Study Design | Setting | Intervention | Study Population | Sample Size | Health Condition | Geographic Area | Data Findings | Main Conclusions Stated by Authors | Quality Rating[b] |
|---|---|---|---|---|---|---|---|---|---|---|
| Stevenson et al., 2021 | Randomized controlled trial | Barbershops | Mental health and violence threat screening | Black men | n = 618 participants | Mental health and violence | Philadelphia, PA | • Intervention had a significant effect on Black manhood vulnerability awareness.<br>• Effects of hypermasculinity and Black manhood vulnerability awareness on the reduction of physical fights were also significant.<br>• Product of the two coefficients on the path from the intervention to the outcome through Black manhood vulnerability awareness was significant, −0.35, 95% ACI [−1.19, −0.02].<br>• Product of the two coefficients on the path from the intervention to the outcome through hypermasculinity was also significant, −0.35, 95% ACI [−1.11, −0.05]. | "This study supports research on how culturally responsive theory-based interventions can increase the likelihood of positive health outcomes in communities of color." | 10/14 |
| Johnson et al., 2010 | Pilot study | Beauty salons | Nutrition education (aimed to increase fruit and vegetable intake) | Black women | n = 20 participants | Poor nutrition, resulting obesity | South Carolina | "The results show that daily servings of fruit and vegetables for the treatment group increased from pretest to posttest, and that fruit and vegetable intake was significantly higher at posttest for the treatment group (P<0.01)." | "The results showed that mean intake of fruit and vegetables was significantly higher at posttest for the treatment group but not for the comparison group. These findings suggest that the intervention may have had a positive effect on fruit and vegetable consumption by treatment group participants." | 8/12 |

*(Continued)*

**Table 3.** (Continued)

| Study | Study Design | Setting | Intervention | Study Population | Sample Size | Health Condition | Geographic Area | Data Findings | Main Conclusions Stated by Authors | Quality Rating[b] |
|---|---|---|---|---|---|---|---|---|---|---|
| Kleindorfer et al., 2008 | Longitudinal study | Beauty salons | Stroke education | Black women | n = 383 participants | Stroke | Cincinnati, OH and Atlanta, GA | • n = 383 completed baseline surveys <br>• n = 318 surveys were completed at 5 months <br>• 78% of women were <60 years old, 69% had some college education, 41% had hypertension, and 12% had diabetes. <br>• Percentage of women who knew 3 warning signs significantly improved from the baseline survey (40.7%) to the final survey (50.6%) <br>• No improvement in knowledge of 3 risk factors (16.5% versus 18.2%). <br>• After our educational intervention, 94% knew to call 911 for stroke symptoms, an 8% improvement over baseline (P = 0.002). | "Despite the challenges of community-based research encountered within our project, we found that stroke education in the beauty shop significantly improved knowledge regarding stroke warning signs and calling 911 among a group of Black women." | 11/14 |
| [a]Madigan et al., 2007 | Cohort study | Barbershops and beauty salons | Chronic kidney disease, hypertension, and diabetes education and connection to resources | Predominantly Black populations | n = 700 trained stylists and n = 14,000 clients | Chronic kidney disease, hypertension, and diabetes | Michigan | • 60% of clients indicating that they have taken steps to prevent diabetes, hypertension, and chronic kidney disease or to seek a physician's advice | "With nearly 60% of clients indicating that they have taken steps to prevent diabetes, hypertension, and chronic kidney disease or to seek a physician's advice, the Healthy Hair program appears to be effective in the short term in prompting attention to healthy behaviors and increasing risk awareness." | 10/14 |
| Leader et al., 2014 | Pilot study | Beauty salons | HPV screening, education, and vaccination promotion | Black women | n = 240 participants | HPV | Philadelphia, PA | "At baseline, 33% of participants answered all of the knowledge questions correctly, while at postintervention, that number rose to 75% and remained at 74% one month later." | "The primary purpose of this pilot study was to assess the feasibility of delivering health education messages to women through the venue of African American beauty salons. The study successfully achieved this goal, while learning significant lessons about how to most effectively deliver such an intervention." | 9/12 |

*(Continued)*

Table 3. (Continued)

| Study | Study Design | Setting | Intervention | Study Population | Sample Size | Health Condition | Geographic Area | Data Findings | Main Conclusions Stated by Authors | Quality Rating [b] |
|---|---|---|---|---|---|---|---|---|---|---|
| Diallo et al., 2023 | Program Evaluation | Barbershops, hair salons, beauty salons, nail salons | Education and resources encouraging COVID-19 vaccination | Northeast Bronx Citizens | n = 45 partner locations (only 27 locations fully completed program) | COVID-19 | Northeast Bronx, New York | "At participating locations in NYC, Bronx, and zip codes 10466, 10469, 10470, and 10475, vaccination rate increases ranged from 5.6% to 8.7% from 1/1/22–5/1/22." | "During the four months of the pilot, zip codes with the highest level of program engagement experienced greater percent increases in COVID-19 vaccination rates during the program period compared to NYC and Bronx averages." | 7/12 |

[a]indicates study is present across multiple tables (Tables 1–3)

[b]Quality assessment of each study design was conducted using the NIH Study Quality Assessment Tools. Full titles of the tools used include: Quality Assessment of Controlled Intervention Studies, Quality Assessment Tool for Observational Cohort and Cross-Sectional Studies, and Quality Assessment Tool for Before-After (Pre-Post) Studies With No Control Group [6]

screenings and reported increased awareness of hypertension and connection to providers by participants.

The other five hypertension studies utilized longitudinal or cluster-randomized trial designs and included baseline and follow-up blood pressure measurements. These studies reported that participants who were connected with pharmacy or other medical providers and provided hypertension education had notable reductions in their blood pressure [10, 11, 13–15]. Two cluster-randomized trials were of particular note due to their study designs (control and intervention groups) and results, which found that the use of barbers as lay health educators and connection with pharmacy and healthcare services led to significant reduction in mean blood pressure in the intervention groups [10, 14]. Two of these hypertension studies with longitudinal components utilized additional unique methods in their interventions. One study incorporated a telehealth component for follow-up care by connecting pharmacy providers with study participants recruited from barbershops online after their blood pressure had been controlled via medication [11]. The other study was a cluster-randomized trial that directly connected some participants to hypertension specialists and observed greater reductions in systolic blood pressure than participants who were first connected to primary care providers [15].

## HIV

A total of 6 studies provided interventions for HIV in this review [19–24]. Sample sizes of these studies ranged from 48–1,124 participants, and the designs of the included studies were a cluster-randomized trial, longitudinal studies, descriptive studies, and qualitative analyses. The majority of HIV intervention studies were conducted in the Northeast region of the United States, with one being conducted in the Southeastern United States [20]. Additionally, while most of the studies primarily targeted predominantly Black populations recruited from laundromats, one study recruited participants from a variety of settings; and another study sampled diverse racial/ethnic participants [20, 23]. All 6 studies provided participants education about HIV and safe sex practices.

One notable study was a descriptive study assessing previous HIV knowledge and providing prevention and education services to Black participants (n = 677) at corner stores, beauty supply stores, laundromats, mechanics, and barbershops [20]. The intervention targeted Black communities heavily affected by HIV in Miami, Florida and included a survey collecting demographic information and HIV knowledge and prevention services like free condom distribution and HIV testing. Within these communities, 68.8% had never heard of PrEP (a medication highly effective at reducing the risk of getting HIV), 8% had never been tested for HIV, and 65.9% had no primary care provider [49]. The study found that the intervention delivered at corner and food stores had the most engagement followed by laundromats, barbershops, and beauty salons.

A second notable intervention study based in Delaware utilized the Popular Opinion Model (POL), a community level peer-based outreach strategy designed by the Centers for Disease Control and Prevention, to provide HIV education to over 6,000 racial/ethnic minority participants and HIV tests to 1,124 of those participants [23].

A third notable HIV study was a cluster randomized controlled trial targeting Black men (n = 618) that compared the Shape Up! Barbers Building Better Brothers HIV risk-reduction intervention (based on the theory of planned behavior) or an attention-matched violence prevention control [24]. The Shape Up! intervention led to a significantly increased consistent condom use in the postintervention period.

These three notable studies together suggest place-based interventions can lead to decreased risky behavior associated HIV transmission (e.g. condomless sex, sex with multiple partners, etc.) and increased self-efficacy for condom use among participants [20, 23, 24].

## Diabetes

A total of 5 studies provided interventions for diabetes were included in the review [8, 12, 18, 25, 26]. Each of these studies occurred at barbershops or beauty salons, and incorporated diabetes education and prevention strategies. The designs of the 5 diabetes studies included a program evaluation, a longitudinal study, a cohort study, a focus group and qualitative research study, and a cross-sectional study. Two of the studies targeted predominantly Black populations, and two additional studies targeted Black men specifically. The remaining study included a broad sample of participants deemed "at risk" for developing diabetes. The sample sizes of the included studies ranged from 13–14,000 participants.

Three studies provided a diabetes intervention in addition to interventions for other conditions [8, 12, 18]. One study was a program evaluation targeting predominantly Black populations (n = 1,823) in barbershops and beauty salons for high blood pressure, diabetes, tobacco-use associated conditions, high cholesterol, and need for social services [8]. Through partnership with an integrated healthcare system, local barbershops and salons in Baltimore, Maryland, and a mobile health clinic, the program screened 469 participants and connected them with free resources when necessary. Another study was a longitudinal study (n = 356) targeting people at risk for diabetes, heart disease, stroke, colon cancer, and breast cancer in laundromats and investigated perceived risk and susceptibility to these conditions; the study found 18% of participants believed their risk for diabetes was lower than it was [12]. The third study was a cohort study that trained over 700 stylists as lay health educators and reported reaching over 14,000 clients with 60% of clients reporting they took steps to prevent or address their diabetes, hypertension, or kidney disease with a provider [18].

The two remaining diabetes studies were a qualitative study and a cross-sectional study, each targeting Black men in barbershops [25, 26]. In the qualitative study, focus groups with 13 participants found that diet and exercise were recognized as ways to prevent diabetes, and people were supportive of barbershops as sites for a diabetes intervention program [25]. The cross-sectional study sampled 290 participants and provided diagnostic hemoglobin A1C testing on site at barbershops and diabetes education based on screening results [26].

## Cancer

A total of 13 studies providing cancer-related prevention, screening, education, and referral services were included in this review [12, 27–38]. Of the 13 studies, 6 of them provided interventions for breast cancer, 4 for prostate cancer, 2 for colorectal cancer, and 1 including both colorectal cancer and breast cancer.

Seven studies provided breast cancer interventions with sample sizes ranging from 162–10,306 participants [12, 32–34, 36–38]. Study designs included longitudinal studies, descriptive studies, randomized-controlled trials, cluster randomized trials, health education programs, and pilot studies. The two cluster randomized trials are noteworthy due to their inclusion of control and intervention groups in their study designs [34, 36]. The majority of studies occurred in beauty salons; but barbershops, churches, neighborhood health centers, laundromats, social service agencies, health fairs, and public libraries were also breast cancer intervention sites. Interventions included health education provided verbally by medical professionals and stylists trained as lay health educators; and education and connection to local resources via kiosks, magazines, store displays, and other paper materials. Regarding target populations,

five of the seven interventions targeted Black women specifically, while the other two targeted participants at risk for breast cancer and all women, respectively [12, 38]. One notable study provided breast cancer education for Black women through touch-screen kiosks located in beauty salons, churches, health fairs, neighborhood health centers, laundromats, public libraries and social service agencies [33]. These kiosks used an interactive computer program called *Reflections of You* that printed magazines for users containing tailored breast cancer education and local breast cancer resources based on participants' answers to screening questions. These kiosks reached 4,527 participants in under 18 months and reported that 34.1% of participants over 40 had never had a mammogram before the intervention. Another descriptive study by the same lead author used the *Reflections of You* kiosks to identify appropriate community channels and settings for delivering evidence-based breast cancer health promotion materials [32]. Through the 10,306 kiosks used over a four year period, the study identified laundromats were the only settings that had the highest kiosk use and highest specificity (e.g. proportion of users without health insurance, barriers to getting a mammogram, low breast cancer and mammography knowledge, etc.).

Four studies provided prostate cancer interventions as two pilot studies, a descriptive study, and a non-randomized comparison study [27, 29, 31, 35]. While three of the four studies targeted Black men in barbershops, one study targeted all men in barbershops, churches, industries, meal sites, car dealerships, civic organizations, and housing projects. Sample sizes of each of these studies ranged from 40–1,552 participants. Each study provided prostate cancer education and prevention materials. One notable study due to its large sample size and unique findings was a descriptive study investigating predictors of participation in free prostate cancer screenings in barbershops, churches, industries, meal sites, car dealerships, civic organizations, and housing projects [31]. The study ultimately found that being white, having at least a high school education, being married, perceiving health benefits, and receiving a client navigator or prior education intervention were significant predictors of participation in the study's free prostate cancer screenings. Another pilot study investigated the feasibility of training barbers to deliver customized (culturally appropriate) prostate cancer education to Black men, mostly through brochures [29]. Through the feasibility pilot study, prostate cancer knowledge scores raised from 60% to 79%.

Three studies provided colorectal cancer interventions in barbershops as a longitudinal study, a qualitative analysis, and a randomized controlled trial respectively [12, 28, 30]. The intervention provided in each of these studies was colorectal cancer education and screening, with two studies focused on Black men and the third study focused on adults in general at risk for developing diabetes, heart disease, stroke, and breast cancer in addition to colorectal cancer. One particularly significant colorectal cancer intervention study due to its incorporation of a telehealth component and large sample size (n = 731) was the randomized controlled trial [30]. This study aimed to test the effectiveness of a preclinical, telephone-based intervention designed to encourage and connect older Black men to colorectal cancer screening opportunities. Black male participants were recruited from barbershops initially and placed in one of three telephone intervention groups: patient navigation by a community health worker for colorectal cancer screening, motivational interviewing by a trained counselor, or both interventions. The study ultimately found that both groups of participants that received navigation by community health worker were most likely to pursue colorectal cancer screening within six months.

## Other conditions

Fourteen studies focused on other conditions not already described above, including high cholesterol, overall physical fitness, mental health, nutrition, stroke, unintended pregnancy,

violence, influenza, kidney disease, HPV, and COVID-19 [8, 12, 18, 39–48]. These studies included cross-sectional studies, longitudinal studies, descriptive studies, feasibility studies, a randomized controlled trial, pilot studies, cohort studies, and program evaluations. Sample sizes ranged from 20–14,000 participants, and target populations included predominantly Black populations, at risk participants for certain conditions, women across all demographic classifications, adults with children 6 months to 2 years of age, adults over 50 years of age, and citizens in the Northeast Bronx region of New York. Each study provided education, prevention strategies, and screening promotion in some capacity for its respective target condition.

These studies took place in barbershops, beauty salons, nail salons, and movie theaters. One descriptive study is particularly noteworthy because of its relatively large sample size (n = 530 participants), specific target population (adults with children 6 months to 2 years of age and adults over 50 years of age), and with the intervention taking place in movie theaters [43]. This intervention was designed to promote annual influenza vaccination by showing slides providing education about the flu and advocating for people to get their annual flu vaccine prior to presentation of upcoming movie premieres. Among moviegoers exposed to the education slides prior to the film, 24% recalled seeing the flu vaccination slides prior to the movie advertisements although some participants did not arrive to the theater before the start of the film to see the flu vaccination slides.

An additional noteworthy study was a randomized controlled trial aimed at assessing the impact of an intervention conducted in barbershops on mental health and violence threat screening among Black men in Philadelphia, PA [44]. With a sample size of 618 participants, the study found significant effects of the intervention on increasing awareness of Black manhood vulnerability. This heightened awareness contributed to a significant reduction in physical fights among participants. The study's findings were robust, demonstrating statistically significant pathways from the intervention through both Black manhood vulnerability awareness and hypermasculinity to the outcomes studied.

Another interesting study investigated the effectiveness of using beauticians to educate Black female clients about stroke warning signs and risk factors [46, 50]. Beauticians were trained about stroke warning signs and risk factors, and clients were asked survey questions about their stroke knowledge before and after the intervention. The study reported significant increases in client knowledge of stroke warning signs (40.7% to 50.6%) and to call 911 for stroke symptoms (86% to 94%) with this improvement sustained for five months. However, no significant increase in knowledge of the three stroke risk factors was seen before and after the intervention.

One final study to note is one focused on providing COVID-19 vaccination education and resources in barbershops, hair salons, beauty salons, and faith-based organizations to Northeast Bronx citizens [48]. Forty-five public-facing business sites across Northeast Bronx, New York participated in this COVID-19 intervention by encouraging clients to complete baseline and follow-up surveys about perceptions of COVID-19 vaccines and commitments to future vaccination, having conversations about COVID-19 and offering supporting materials, displaying posters and brochures on site encouraging vaccination, and hosting local health department staff on site. Over a span of four months, COVID-19 vaccination rates across five zip codes in Northeast Bronx were observed to increase from 5.6% to 8.7%, although causality of the intervention cannot be inferred.

## Discussion

This unique review of place-based health interventions in public business settings found that a number of studies have been conducted on the topic in the past two decades. We reviewed 42

studies of place-based health interventions offered for various chronic health conditions and certain select business settings. The majority of the interventions offered were health education and preventative health screenings. Thirty-four (81%) of the studies focused on reaching Black populations; all studies, except for one, delivered health interventions in barbershops and beauty salons either solely or among a few other settings. The largest number of studies focused on cancer (13 studies) or cardiovascular disease (12 studies). Additionally, the specific health condition with the greatest number of controlled trials was for hypertension (3 studies).

In general, studies reported that health interventions embedded in public settings were associated with positive outcomes, including increased disease awareness, improved health behaviors and disease management, and high rates of health screening and connection to healthcare services. Given that all studies targeted outreach to racial/ethnic minority populations, the findings suggest place-based interventions are an important way to reach underserved population and potentially address health disparities by providing accessible health education, screenings, and connection to services in places that they visit to purchase goods and services. However, it is essential to acknowledge that conducting health interventions in public spaces may inadvertently lead to unintended consequences, such as stigma and concerns about privacy. The presence of stigma or concerns about privacy can affect participant engagement and willingness to utilize these services, potentially undermining the effectiveness of the interventions. Thus, careful consideration of these social dynamics is crucial when designing and implementing interventions in community settings. Placements of these interventions in barbershops and beauty salons may represent familiar and trusted community settings that can enhance participant engagement. Further study is needed to expand beyond these settings to determine whether other public-facing business settings (e.g., banks, movie theatres, malls) are also effective and acceptable places for health interventions.

Among the 42 studies reviewed, there was a general lack of rigor in the designed studies. Only one study received the maximum quality rating score [24], and no other study was within 2 point of the maximum quality rating score. Most studies were descriptive or observational one-group designs and did not include a comparison group. Although we identified 7 randomized trials, including several cluster randomized trials, these studies varied widely in terms of interventions and health conditions. As a result, we did not attempt to synthesize the results quantitatively. Instead, we summarized the findings of each study individually, highlighting their diverse approaches and outcomes. This decision was based on the heterogeneity of the studies, which made direct comparison challenging. In addition, many of these trials did not appear to be rigorously designed and may have had many threats to internal validity (e.g., confounding variables, inadequate sample size, limited follow-up or differential attrition between groups) that were not fully examined. Almost all studies relied on subjective outcome measures and did not measure objective health outcomes (e.g., service utilization, disease onset and outcome) so there is a need for further rigorous studies with objective outcomes. Together, our review concludes there is a small, growing body of studies of health interventions delivered in public-facing business settings that shows some preliminary success in reaching Black communities for a variety of health conditions, and these interventions may be a promising strategy to reach underserved populations but more rigorous and varied studies are needed to expand and deepen the evidence for these interventions to pinpoint how they are effective, who they are most effective for, and in using which interventions in what places.

This review had several strengths and limitations worth noting. Given the nature of scoping reviews, we took a broad, comprehensive approach to cover a wide range of health conditions, interventions, settings, and study designs. There was wide variability in studies making it challenging to compare studies, and a meta-analysis could not be conducted to quantify a summary of outcomes. Given the range of studies, we are also limited in specificity in drawing

conclusions. However, we have tried to summarize findings by health condition to organize the studies and allow researchers to focus on particular health conditions. We only included studies in the United States, and there may be various innovative place-based interventions delivered internationally in other countries that would yield new insights so that is both a limitation of our review and an opportunity for future research. Moreover, we only include published studies, and there may be a "file-drawer problem" of unpublished studies we do not include. Finally, while stakeholder consultation is recognized as beneficial in scoping reviews, no formal stakeholder consultations were conducted for this review, which we acknowledge as a limitation. Additionally, while our approach aligns with best practices for scoping reviews, the absence of protocol registration may limit transparency. We also recognize this as a potential limitation to reproducibility.

These limitations notwithstanding, this review highlights unique and innovative ways to reach underserved populations in places like barbershops and beauty salons. The strongest evidence for these place-based interventions is for cardiovascular disease (especially hypertension) and cancer, but there are opportunities to study this further for various other health conditions. Together, these studies demonstrate possible collaborations between healthcare providers, researchers, and business owners with mutual goals to serve underserved communities. Finally, this review paves numerous paths for needed research in this area, including more experimental studies with objective outcomes, examination of the sustainability and scalability of these interventions, and the cost-effectiveness of interventions to support their adoption by businesses, healthcare providers, and policymakers.

## Supporting information

**S1 Table. Keywords/operators/truncation used in databases for systematic review.**
(DOCX)

**S1 Checklist. PRISMA 2020 checklist.**
(DOCX)

## Author Contributions

**Conceptualization:** Jack Tsai, Nicholas A. McCann.

**Data curation:** Nicholas A. McCann.

**Formal analysis:** Nicholas A. McCann.

**Investigation:** Nicholas A. McCann.

**Methodology:** Nicholas A. McCann.

**Project administration:** Jack Tsai.

**Software:** Nicholas A. McCann.

**Supervision:** Jack Tsai.

**Validation:** Nicholas A. McCann.

**Visualization:** Nicholas A. McCann.

**Writing – original draft:** Jack Tsai, Nicholas A. McCann.

**Writing – review & editing:** Jack Tsai, Nicholas A. McCann.

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
