## [Decision Letter · Decision Letter 0]

27 Sep 2024

PONE-D-24-30536Usual prevention in unusual settings: A scoping review of place-based health interventions in public-facing businessesPLOS ONE

Dear Dr. Tsai,

Thank you for submitting your manuscript to PLOS ONE. After careful consideration, we feel that it has merit but does not fully meet PLOS ONE’s publication criteria as it currently stands. Therefore, we invite you to submit a revised version of the manuscript that addresses the points raised during the review process.

We look forward to receiving your revised manuscript.

Kind regards,

Lakshminarayana Chekuri, MD, PhD

Academic Editor

PLOS ONE

Journal Requirements: When submitting your revision, we need you to address these additional requirements. 1. Please ensure that your manuscript meets PLOS ONE's style requirements, including those for file naming. The PLOS ONE style templates can be found at https://journals.plos.org/plosone/s/file?id=wjVg/PLOSOne_formatting_sample_main_body.pdf and https://journals.plos.org/plosone/s/file?id=ba62/PLOSOne_formatting_sample_title_authors_affiliations.pdf 2. We note that your Data Availability Statement is currently as follows: All relevant data are within the manuscript and its Supporting Information files. Please confirm at this time whether or not your submission contains all raw data required to replicate the results of your study. Authors must share the “minimal data set” for their submission. PLOS defines the minimal data set to consist of the data required to replicate all study findings reported in the article, as well as related metadata and methods (https://journals.plos.org/plosone/s/data-availability#loc-minimal-data-set-definition). For example, authors should submit the following data: - The values behind the means, standard deviations and other measures reported;- The values used to build graphs;- The points extracted from images for analysis. Authors do not need to submit their entire data set if only a portion of the data was used in the reported study. If your submission does not contain these data, please either upload them as Supporting Information files or deposit them to a stable, public repository and provide us with the relevant URLs, DOIs, or accession numbers. For a list of recommended repositories, please see https://journals.plos.org/plosone/s/recommended-repositories. If there are ethical or legal restrictions on sharing a de-identified data set, please explain them in detail (e.g., data contain potentially sensitive information, data are owned by a third-party organization, etc.) and who has imposed them (e.g., an ethics committee). Please also provide contact information for a data access committee, ethics committee, or other institutional body to which data requests may be sent. If data are owned by a third party, please indicate how others may request data access. 3. We note that you have included the phrase “data not shown” in your manuscript. Unfortunately, this does not meet our data sharing requirements. PLOS does not permit references to inaccessible data. We require that authors provide all relevant data within the paper, Supporting Information files, or in an acceptable, public repository. Please add a citation to support this phrase or upload the data that corresponds with these findings to a stable repository (such as Figshare or Dryad) and provide and URLs, DOIs, or accession numbers that may be used to access these data. Or, if the data are not a core part of the research being presented in your study, we ask that you remove the phrase that refers to these data.

**Additional Editor Comments:**

Thank you for your scholarly contribution. I'd also like to thank the authors for choosing PLOS ONE to publish your findings from this study. Comments from reviewers are provided below. Please review these comments and I suggest address them and resubmit your manuscript. Your timely response would help this study be published and accessible to interested readers across the world. I look forward to reviewing your revised manuscript. I wish you good luck with your future endeavors.

Reviewers' comments:

Reviewer's Responses to Questions

**Comments to the Author**

1. Is the manuscript technically sound, and do the data support the conclusions?

Reviewer #1: Partly

Reviewer #2: Partly

2. Has the statistical analysis been performed appropriately and rigorously? 

Reviewer #1: N/A

Reviewer #2: Yes

3. Have the authors made all data underlying the findings in their manuscript fully available?

Reviewer #1: Yes

Reviewer #2: Yes

4. Is the manuscript presented in an intelligible fashion and written in standard English?

Reviewer #1: Yes

Reviewer #2: Yes

5. Review Comments to the Author

Reviewer #1: This article presents a scoping review of peer reviewed articles on the broad topic of place-based health interventions. They present the findings by health issue of 43 peer reviewed articles and provide information on geographic location, sample population and size, and main findings. They found that barber shops and beauty salons were common facilities for cardiovascular disease and cancer interventions (i.e., screenings, referrals, and education), particularly in populations that were African American or Black.

I see a couple of major concerns about the synthesis of the findings and a need for transparency of the methods to determine rigor. These concerns prevent me from endorsing its acceptance in its present state.

The study fails to clarify how this scoping review builds upon a recent umbrella review by McGowan et al (2021). In addition, this article’s main finding of use of barbershops and beauty salons for public health outreach, research, and health intervention in racial and ethnic minority populations is already well established. For example, W. Kong and E. Saunders published in this context about hypertension in 1989. Additional synthesis of data by health issue, geographic region, strength of evidence, etc. would be helpful to the reader. For example, an evidence map may be relevant.

I have concerns about the level of sufficient detail to determine whether the scoping review was performed with rigor. Increased transparency in the methods is essential. The authors are off to a good start by using the PRISMA checklist for reporting. However, several key areas are not included or are included with insufficient detail. There was only one reviewer and yet no information about attempts to reduce bias. They should thoroughly address failure to implement best practices from Mak and Thomas (2022). For example, the review protocol should be registered and the management software named. They should describe methods used to synthesize results. They should describe the process of extraction, calibration, stakeholder consultation, and beyond. The lack of assessment on risk of bias or analysis of strength of evidence should be explored as described in the PRISMA checklist.

As part of the discussion, in exploring these findings, the unintended consequences (ie. stigma) should be noted in conducting any health intervention in a public space. Experience with community advisory boards helps tailor interventions with cultural competence. In addition, greater discussion of the limitations and gaps is needed.

Specific minor feedback for consideration is included below:

• Figure 1 is very challenging to read despite zooming in. Make the text larger and increase the quality of the image. Additional details to clarify the process should be included.

• “Black” should be capitalized consistently throughout when referring to groups in racial, cultural, or ethnic terms.

• Justification is needed on why only peer reviewed articles were included and what impact that could have on the findings.

• The full syntax of keywords and strategies used in the initial searches should be available as supplemental materials, not available upon request from the authors.

References:

Mak S, Thomas A. Steps for Conducting a Scoping Review. J Grad Med Educ. 2022 Oct;14(5):565-567.

McGowan, V.J., Buckner, S., Mead, R. et al. Examining the effectiveness of place-based interventions to improve public health and reduce health inequalities: an umbrella review. BMC Public Health 21, 1888 (2021).

Kong W., Saunders E. (1989). Community programs to increase hypertension control. Journal of the National Medical Association, 81, 13–16.

Reviewer #2: Comments to Authors

Overall Feedback

The manuscript presents an important review on the use of non-traditional business settings for health interventions, which is both timely and relevant. I believe that addressing the structural and methodological concerns will significantly strengthen the manuscript.

Abstract

• Please provide a clearly defined and focused objective. The current version is too broad, making it difficult for the reader to gauge the expectations and scope of the review. I recommend using the objective stated in the Introduction for consistency and clarity.

• Additionally, a clarification is needed for the statement: “While there have been 9 randomized trials conducted across health conditions, they have been limited to a few select settings with little measurement of objective health outcomes indicating a need for more rigorously designed studies.” It is unclear whether the concern is the limited rigor of the randomized trials or their restriction to select settings with inadequate focus on objective health outcomes. I will suggest that authors address this ambiguity for a clearer interpretation.

Introduction

The introduction is well-structured but could be improved further.

• The statement, “However, these interventions were mostly focused on altering the ‘place’ of people as the intervention rather than focusing on delivering usual interventions in unusual places or settings,” requires expansion. It is not immediately apparent why altering the "place" of people is less favorable compared to delivering interventions in unconventional settings. Further explanation of these approaches and their respective merits or shortcomings would add depth to the argument.

• The example cited “For example, there have been a number of studies that have developed health promotion interventions in salons and barbershops to reach African American communities and a systematic review has summarized this research [3]” is a good one, but it feels disconnected from the preceding sentences. I will suggest re-writing the paragraph to provide a more coherent narrative, clearly linking examples to the argument.

• There is also a lack of an explicit explanation of how this review fills a gap in the literature. Currently, the significance of focusing on "unusual business settings" for health interventions is not adequately emphasized. Further explanation on why these settings are essential for addressing health disparities or improving outreach would strenghten the introduction. The objective should also address the "so what?" question to clarify the overall goal of the review.

Methods

There are some concerns in this section that need to be addressed.

• First, the manuscript states that "four major search databases were used: PubMed, Google Scholar, and APA PsycNet," but this lists only three databases.

• Also, the rationale behind limiting the search to studies from 1995-2023 is not provided. An explanation for this temporal restriction is needed to justify its relevance to the review.

• The search syntax appears overly simplistic. I recommend following established search strategies from previous reviews, such as those by McGowan et al. (2021) and Palmer et al. (2021), which may provide more robust search terms. In addition, the search seems to focus exclusively on barbershops, laundromats, hair salons, movie theaters, and nail salons, but there is no explanation for excluding other business settings that might serve as intervention sites. Expanding this scope, or at least justifying the exclusion of other settings, would improve the comprehensiveness of the review.

• It is best practice to include both inclusion and exclusion criteria explicitly, which are missing in this section. Additionally, since the review appears to be conducted by a single author, this raises concerns about potential bias. I recommend that co-authors participate in the data extraction process and that at least two reviewers are involved to verify extracted data and ensure consistency, which will strengthen the rigor of the review.

Results

• The current organization of the results does not align well with the review’s objectives. I recommend thematizing the findings based on the various types of preventive health interventions rather than organizing them by health conditions. This would allow for a more structured discussion of intervention methods, such as screenings for hypertension, HIV, or diabetes, while also categorizing them into primary, secondary, and tertiary prevention levels.

• For these sentences, “Table 1 contained studies providing interventions for cardiovascular health conditions, HIV, and diabetes. Table 2 contained studies providing interventions for different cancers, and Table 3 contained studies providing interventions for all other conditions not falling.”, Please cite the studies that belong to each of these categories. For instance, when discussing cardiovascular interventions, specific studies should be cited for clarity.

• The sentence, "These studies included screening programs, cross-sectional studies, longitudinal studies, descriptive studies, feasibility studies, a randomized controlled trial, pilot studies, cohort studies, and program evaluations," seems unclear in its inclusion of screening programs. Is there a specific reason screening programs are listed alongside study designs? Clarifying this distinction would improve clarity.

• Including more commentary on intervention outcomes, particularly which intervention methods were most effective, would add depth to the analysis.

• Adding more commentary on intervention outcomes, particularly on which methods were the most effective, would enrich the analysis and provide readers with a clearer sense of the practical impact of these interventions.

• More detail on the effectiveness of the interventions is needed, especially in terms of measurable health outcomes. For instance, while the review notes positive participant responses, including data on health improvements or behavior changes (e.g., increases in screenings or follow-up rates) would strengthen the results section.

• The tables are well-structured, but adding a column that summarizes the limitations of each study—such as small sample sizes or the observational nature of some interventions—would be helpful, especially since the authors are interested in the rigor of study designs.

Discussion

• The discussion section should demonstrate stronger alignment with the review's objective. For instance, the statement "we reviewed 43 studies of place-based health interventions" should clearly connect to the objective, which is to assess interventions delivered in non-traditional settings.

• “Additionally, the specific health condition with the greatest number of rigorous research designs (e.g. randomized controlled trials, cluster randomized trials) was for hypertension (3 studies).” I do not agree with this statement. The claim that randomized controlled trials are more rigorous than other study designs is problematic. The rigor of a study design depends on its alignment with the research question. Cross-sectional studies can be as rigorous as randomized controlled trials if they are appropriately suited to the research aims.

• “Further study is needed to expand beyond these settings to determine whether other public-facing business settings (e.g., grocery stores, banks, movie theatres, malls) are also effective and acceptable places for health interventions.” Did any of the reviewed studies focus on these settings or is there a reason why authors did not review articles from these settings?

• The sentence, "Although we counted 7 randomized trials, including several cluster randomized trials, they offered different types of interventions and targeted different health conditions, making synthesis challenging," is unclear. Could the authors clarify whether they attempted to synthesize information from these studies? If not, what approach did they take to handle these trials?

• "Many of these trials appeared to have design flaws, potentially compromising internal validity due to factors such as confounding variables, inadequate sample sizes, limited follow-up, or differential attrition between groups." Since the authors are concerned with study design rigor, it would be beneficial to evaluate the quality of each trial using established guidelines.

• The statement "Almost all studies relied on subjective outcome measures and did not measure..." is incomplete and not discussed in the results section. The authors should include this in the results if it is an important observation.

• Given the range of studies, we are also limited in specificity in drawing conclusions” Authors should please clarify what this statement means.

• "We only included studies from the United States, but there may be innovative place-based interventions in other countries" requires justification in the introduction. Why was the focus restricted to U.S.-based studies?

• While the review highlights innovative methods for reaching underserved populations in barbershops and beauty salons, the current structure of the results may not fully support this claim. Reorganizing the results by types of place-based health intervention first, followed by specific interventions, and then health conditions, would provide a clearer framework for discussion.

• A more detailed exploration of the barriers encountered in implementing health interventions in these settings should be included in the discussion. Challenges related to business owners’ willingness to participate, or logistical issues should be addressed.

• Expanding the discussion to include policy implications for promoting place-based health interventions, particularly in underserved communities, would add valuable context to the findings.

• Additionally, reflecting on how the review's findings could guide future research on health disparities would strengthen the overall contribution of the paper.

6. PLOS authors have the option to publish the peer review history of their article (what does this mean?). If published, this will include your full peer review and any attached files.

Reviewer #1: No

Reviewer #2: No

---

## [Author Response · Author response to Decision Letter 0]

20 Nov 2024

Dear Dr. Chekuri and Reviewers

Thank you for the opportunity to revise and resubmit our article entitled “Usual prevention in unusual settings: A scoping review of place-based health interventions in public-facing businesses” (PONE-D-24-30536). We appreciate the Reviewers’ constructive comments and have worked to address each of them, which we detail below. We believe the review has been strengthened in the process and hope you can reconsider this work for publication in PLOS ONE. Thank you greatly for your time.

For Reviewer #1:

1. “The study fails to clarify how this scoping review builds upon a recent umbrella review by McGowan et al (2021). In addition, this article’s main finding of use of barbershops and beauty salons for public health outreach, research, and health intervention in racial and ethnic minority populations is already well established. For example, W. Kong and E. Saunders published in this context about hypertension in 1989. Additional synthesis of data by health issue, geographic region, strength of evidence, etc. would be helpful to the reader. For example, an evidence map may be relevant.”

Response #1: We appreciate the reviewer’s suggestion to clarify how this review builds on existing literature, such as the recent umbrella review by McGowan et al. (2021). In response, we have expanded the introduction to clarify that, unlike prior reviews that largely focus on interventions in more conventional settings, our scoping review examines preventive health interventions in a broader array of unconventional business settings. We have revised the text on pages 2-3 to emphasize this distinction and highlight our review’s unique focus on public-facing venues outside of commonly studied sites such as grocery stores and pharmacies.

2. “I have concerns about the level of sufficient detail to determine whether the scoping review was performed with rigor. Increased transparency in the methods is essential. The authors are off to a good start by using the PRISMA checklist for reporting. However, several key areas are not included or are included with insufficient detail. There was only one reviewer and yet no information about attempts to reduce bias. They should thoroughly address failure to implement best practices from Mak and Thomas (2022). For example, the review protocol should be registered and the management software named. They should describe methods used to synthesize results. They should describe the process of extraction, calibration, stakeholder consultation, and beyond. The lack of assessment on risk of bias or analysis of strength of evidence should be explored as described in the PRISMA checklist.”

Response #2: Thank you for your feedback on ensuring sufficient rigor in our methods. We have revised the manuscript to clarify that two reviewers participated in all stages of the review, including with one reviewer leading study screening, selection, and data extraction and checked by the second reviewer with corrections made when discrepancies were found. We provided additional detail on our calibration process, where both reviewers independently extracted data from a sample of studies to ensure consistency. We also clarified our management approach, noting the use of Excel for data collection and Zotero for citation management. While the review protocol was not pre-registered, we aligned our methods with PRISMA guidelines for scoping reviews. We acknowledged the lack of formal stakeholder consultation as a limitation. Additionally, we described our results synthesis by categorizing studies according to intervention type, health condition, and setting. Finally, we assessed the methodological quality of each study using the NIH Study Quality Assessment Tools to provide insights into the strength of the evidence presented. These changes were implemented on pages 3-6 and 18 to enhance the transparency and rigor of our scoping review.

3. “As part of the discussion, in exploring these findings, the unintended consequences (ie. stigma) should be noted in conducting any health intervention in a public space. Experience with community advisory boards helps tailor interventions with cultural competence. In addition, greater discussion of the limitations and gaps is needed.”

Response #3: Thank you for your valuable feedback regarding the need to address stigma in health interventions. We have revised the manuscript to include this important aspect. On page 17, we now discuss how health interventions in public settings may lead to stigma, potentially affecting participant engagement. We highlight the importance of understanding these dynamics to ensure interventions are culturally competent and effectively tailored to the communities served. This addition strengthens our discussion and emphasizes the need to consider unintended consequences in place-based health interventions.

4. “Figure 1 is very challenging to read despite zooming in. Make the text larger and increase the quality of the image. Additional details to clarify the process should be included.”

Response #4: We have now uploaded an original image for Figure 1 which should be higher resolution. 

5. “’Black’ should be capitalized consistently throughout when referring to groups in racial, cultural, or ethnic terms.”

Response #5: Thank you for your valuable feedback regarding the capitalization of "Black" in our manuscript. We have revised the text to ensure that "Black" is now consistently capitalized throughout when referring to groups in racial, cultural, or ethnic terms. We appreciate your attention to this detail, as it enhances the accuracy and professionalism of our writing.

6. “Justification is needed on why only peer reviewed articles were included and what impact that could have on the findings.”

Response #6: Thank you for your insightful comment regarding the inclusion of only peer-reviewed articles in our review. We chose to limit our analysis to peer-reviewed studies to ensure the rigor and credibility of the research findings. Peer review serves as a quality control mechanism that helps validate the methodology and results of a study, thereby enhancing the reliability of the evidence presented. By focusing on peer-reviewed literature, we aimed to provide a more robust synthesis of place-based health interventions, ensuring that the interventions evaluated have undergone critical scrutiny by experts in the field. We acknowledge that this choice may limit the scope of included studies and could exclude valuable insights from non-peer-reviewed literature. However, we believe that prioritizing peer-reviewed research contributes to the overall strength and validity of our review's conclusions. We now provide a justification in the Methods section.

7. “The full syntax of keywords and strategies used in the initial searches should be available as supplemental materials, not available upon request from the authors.”

Response #7: We now provide a full list of syntax used as an Appendix item. 

For Reviewer #2:

Abstract

1. “Please provide a clearly defined and focused objective. The current version is too broad, making it difficult for the reader to gauge the expectations and scope of the review. I recommend using the objective stated in the Introduction for consistency and clarity.”

Response #1: Thank you for your insightful comment regarding the need for a clearer and more focused objective. In response, we have revised the objective in the abstract to better articulate the scope and expectations of the review. The objective now clearly states the aim of summarizing the peer-reviewed literature on place-based health interventions in unconventional public-facing business settings, specifically highlighting the target populations and health conditions addressed. These changes can be found on page 1 of the revised manuscript.

2. “Additionally, a clarification is needed for the statement: “While there have been 9 randomized trials conducted across health conditions, they have been limited to a few select settings with little measurement of objective health outcomes indicating a need for more rigorously designed studies.” It is unclear whether the concern is the limited rigor of the randomized trials or their restriction to select settings with inadequate focus on objective health outcomes. I will suggest that authors address this ambiguity for a clearer interpretation.”

Response #2: Thank you for your insightful feedback regarding the statement on the randomized trials. To address the ambiguity, we have revised the text for clarity. The updated sentence now explicitly distinguishes the two concerns regarding the randomized trials: their restriction to select settings and the limited rigor in their design, particularly concerning the measurement of objective health outcomes. The revised statement now more simply and clearly reads: “While there have been 9 randomized trials conducted across various health conditions, these trials are limited in two key ways: first, they focus on a few select settings and lack of objective health out. This highlights the need for more rigorously designed studies in diverse settings that can effectively evaluate the impact of place-based health interventions.” These changes were made on page 1. We appreciate your guidance in enhancing the clarity of our manuscript.

Introduction:

3. “The statement, “However, these interventions were mostly focused on altering the ‘place’ of people as the intervention rather than focusing on delivering usual interventions in unusual places or settings,” requires expansion. It is not immediately apparent why altering the "place" of people is less favorable compared to delivering interventions in unconventional settings. Further explanation of these approaches and their respective merits or shortcomings would add depth to the argument.”

Response #3: We appreciate your feedback regarding the need for expansion on the statement concerning the focus on altering the "place" of people versus delivering usual interventions in unconventional settings. In response, we have clarified this point in the paragraph by emphasizing that altering the “place” of people can disconnect individuals from their familiar community contexts. We argue that delivering traditional health interventions in unconventional settings—such as salons and barbershops—preserves the familiarity and trust these spaces offer, which can lead to better engagement and participation from target populations. This addition enriches the discussion by providing a clearer rationale for why the context of the intervention matters. These changes were made on page 2.

4. “The example cited “For example, there have been a number of studies that have developed health promotion interventions in salons and barbershops to reach African American communities and a systematic review has summarized this research [3]” is a good one, but it feels disconnected from the preceding sentences. I will suggest re-writing the paragraph to provide a more coherent narrative, clearly linking examples to the argument.”

Response #4: Thank you for your feedback regarding the coherence of the example within the paragraph. We have revised the paragraph to create a clearer and more cohesive narrative that better links the example to the overarching argument. Specifically, we clarified how altering the "place" of people can disconnect them from familiar community settings, in contrast to delivering interventions in unconventional but trusted spaces like salons and barbershops. This revision illustrates how using such settings can enhance engagement and accessibility for target populations. These changes were made to improve the flow and strengthen the argument within the paragraph on pages 2-3.

5. “There is also a lack of an explicit explanation of how this review fills a gap in the literature. Currently, the significance of focusing on "unusual business settings" for health interventions is not adequately emphasized. Further explanation on why these settings are essential for addressing health disparities or improving outreach would strengthen the introduction. The objective should also address the "so what?" question to clarify the overall goal of the review.”

Response #5: Thank you for highlighting the need for a clearer explanation of this review’s significance. In response, we have revised the introduction to explicitly state how focusing on unconventional business settings for health interventions addresses a gap in the literature. We emphasize that these settings—such as salons, barbershops, and other non-traditional venues—provide a unique opportunity to reach underserved populations in familiar, trusted spaces, potentially enhancing engagement and access to health resources. We have also clarified the objective to highlight how these unconventional settings can contribute to addressing health disparities and extending outreach. These revisions were made to ensure that the purpose and importance of the review are clear, providing a stronger rationale for its focus.

Methods:

6. “First, the manuscript states that "four major search databases were used: PubMed, Google Scholar, and APA PsycNet," but this lists only three databases.”

Response #6: Thank you for catching this inconsistency. We have corrected the text to reflect that three major search databases were used (PubMed, Google Scholar, and APA PsycNet) instead of four. This change has been made on page 4.

7. “Also, the rationale behind limiting the search to studies from 1995-2023 is not provided. An explanation for this temporal restriction is needed to justify its relevance to the review.”

Response #7: Thank you for your comment. We have clarified in the manuscript that while the search range in each database was set from 1960 to the present, the studies that met our inclusion criteria were published between 1995 and 2023. This revision has been made on page 4.

8. “The search syntax appears overly simplistic. I recommend following established search strategies from previous reviews, such as those by McGowan et al. (2021) and Palmer et al. (2021), which may provide more robust search terms. In addition, the search seems to focus exclusively on barbershops, laundromats, hair salons, movie theaters, and nail salons, but there is no explanation for excluding other business settings that might serve as intervention sites. Expanding this scope, or at least justifying the exclusion of other settings, would improve the comprehensiveness of the review.”

Response #8: Thank you for this feedback. We have now included the exact search terms and syntax used in our search, which included the strategies suggested. We focused on sites, such as barbershops, laundromats, hair salons, movie theaters, and nail salons—that met our definition of "unconventional" business settings where healthcare interventions had not usually be considered historically. These settings were selected based on their established role in the community and their alignment with our review’s objective of identifying less traditional sites for health interventions. Expanding to additional settings was beyond the scope of this review; however, we agree that exploring a broader range of business types could be valuable for future research.

9. “It is best practice to include both inclusion and exclusion criteria explicitly, which are missing in this section. Additionally, since the review appears to be conducted by a single author, this raises concerns about potential bias. I recommend that co-authors participate in the data extraction process and that at least two reviewers are involved to verify extracted data and ensure consistency, which will strengthen the rigor of the review.”

Response #9: Thanks for this important point. We have now clarified both inclusion and exclusion criteria explicitly. In short, inclusion criteria were “studies conducted in the United States, provided health interventions (e.g., education, disease screening, connection with healthcare providers, pharmacy services) to clients in one or more business settings for one or more diseases or conditions, were written in English, and were peer-reviewed.” And exclusion criteria were “studies that were not peer-reviewed, not p

---

## [Decision Letter · Decision Letter 1]

16 Dec 2024

PONE-D-24-30536R1Usual prevention in unusual settings: A scoping review of place-based health interventions in public-facing businessesPLOS ONE

Dear Dr. Tsai,

Thank you for submitting your manuscript to PLOS ONE. After careful consideration, we feel that it has merit but does not fully meet PLOS ONE’s publication criteria as it currently stands. Therefore, we invite you to submit a revised version of the manuscript that addresses the points raised during the review process.

We look forward to receiving your revised manuscript.

Kind regards,

Lakshminarayana Chekuri, MD, PhD

Academic Editor

PLOS ONE

Journal Requirements:

Additional Editor Comments :

Please review attached document for my suggestions.

Reviewers' comments:

Reviewer's Responses to Questions

**Comments to the Author**

1. If the authors have adequately addressed your comments raised in a previous round of review and you feel that this manuscript is now acceptable for publication, you may indicate that here to bypass the “Comments to the Author” section, enter your conflict of interest statement in the “Confidential to Editor” section, and submit your "Accept" recommendation.

Reviewer #1: (No Response)

Reviewer #2: All comments have been addressed

2. Is the manuscript technically sound, and do the data support the conclusions?

Reviewer #1: Yes

Reviewer #2: Yes

3. Has the statistical analysis been performed appropriately and rigorously? 

Reviewer #1: N/A

Reviewer #2: N/A

4. Have the authors made all data underlying the findings in their manuscript fully available?

Reviewer #1: Yes

Reviewer #2: Yes

5. Is the manuscript presented in an intelligible fashion and written in standard English?

Reviewer #1: Yes

Reviewer #2: Yes

6. Review Comments to the Author

Reviewer #1: I appreciate the numerous changes made by the authors to address the reviewers’ concerns. These improved the transparency in the rigor of the methods.

My previous concern, as well as that noted by Reviewer #2, was partially addressed; how does this review build upon the existing literature (i.e. McGowan et al 2021)? As previously noted, this article’s main finding of use of barbershops and beauty salons for public health outreach, research, and health intervention in racial and ethnic minority populations is already well-established. The author's clarified that organizing their findings by health issue adds value to the field as it synthesizes the evidence and quality to plan for future research and intervention.

Please clarify the revisions the authors’ noted on page 2 as they present two unclear and conflicting sentences (below).

“For example, there have been a number of studies that have developed health promotion interventions in salons and barbershops to reach African American communities, with a systematic review summarizing this research [3]. However, these studies have primarily focused on settings traditionally associated with community health outreach, such as grocery stores, pharmacies, and similar venues.”

Reviewer #2: The authors have done a great job addressing the concerns I raised in the first round of review. I have no additional comments. Well done.

7. PLOS authors have the option to publish the peer review history of their article (what does this mean?). If published, this will include your full peer review and any attached files.

Reviewer #1: No

Reviewer #2: No

---

## [Author Response · Author response to Decision Letter 1]

17 Dec 2024

Dear Dr. Chekuri and Reviewers

Thank you for the opportunity to revise and resubmit our article entitled “Usual prevention in unusual settings: A scoping review of place-based health interventions in public-facing businesses” (PONE-D-24-30536R1). We appreciate the Editor’s careful reading and suggested edits along with each of the Reviewers’ positive and encouraging comments.

In this revised submission, we have now made all of the Editor’s suggested edits throughout the manuscript. For Reviewer #2, we have clarified the two sentences (bottom of page 2 and top of page 3) that there have been a few novel studies of interventions in unconventional settings like barbershops, but the majority of studies and the work in real-world settings continue to be in traditional settings like pharmacies and community health centers. We hope these revisions have improved the manuscript.

We greatly appreciate the opportunity to resubmit our work and hope this revised manuscript can be considered for publication. Thank you for your time.

---

## [Editor Report · Decision Letter 2]

27 Dec 2024

PONE-D-24-30536R2Usual prevention in unusual settings: A scoping review of place-based health interventions in public-facing businessesPLOS ONE

Dear Dr. Tsai,

Thank you for submitting your manuscript to PLOS ONE. After careful consideration, we feel that it has merit but does not fully meet PLOS ONE’s publication criteria as it currently stands. Therefore, we invite you to submit a revised version of the manuscript that addresses the points raised during the review process.

We look forward to receiving your revised manuscript.

Kind regards,

Lakshminarayana Chekuri, MD, PhD

Academic Editor

PLOS ONE

Journal Requirements:

Additional Editor Comments:

Please review my comments in attached document.

---

## [Author Response · Author response to Decision Letter 2]

30 Dec 2024

Dear Dr. Chekuri,

Thank you for the opportunity to make some final corrections to the manuscript. Your comments were very helpful, and we have made each of the following revisions.

In the Abstract on Page 1, we have revised the sentence to now read: “The most common unconventional public-facing business settings for health interventions included barbershops or beauty salons…” We took out the number of studies because we agree it’s confusing and potentially misleading. You are also very correct that it is 34 (81%) of studies targeted Black populations. We have corrected this in the Abstract and in the Discussion, thank you very much.

On Page 2, we have revised the sentence as suggested to read: “Place-based interventions is one approach to reach people ‘where they are.’”

On Page 3, we have corrected a number of typos and reworded several sentences as you suggested. Thank you for these great suggestions.

On Page 5, we have taken out the word “reviewer.” On Page 7, we have corrected the wording to now read “…each with a sample size of 37 participants…”

On Page 18, we have corrected the sentence to now read “…there was a general lack of rigor in the designed studies." Thank you.

---

## [Editor Report · Decision Letter 3]

7 Jan 2025

Usual prevention in unusual settings: A scoping review of place-based health interventions in public-facing businesses

PONE-D-24-30536R3

Dear Dr. Tsai,

We’re pleased to inform you that your manuscript has been judged scientifically suitable for publication and will be formally accepted for publication once it meets all outstanding technical requirements.

Kind regards,

Lakshminarayana Chekuri, MD, PhD

Academic Editor

PLOS ONE
---

## [Editor Report · Acceptance letter]

8 Jan 2025

PONE-D-24-30536R3 

PLOS ONE

Dear Dr. Tsai, 

I'm pleased to inform you that your manuscript has been deemed suitable for publication in PLOS ONE. Congratulations! Your manuscript is now being handed over to our production team.

Kind regards, 

on behalf of

Dr. Lakshminarayana Chekuri 

Academic Editor

PLOS ONE
